# RefDrop: Controllable Consistency in Image or Video Generation via Reference Feature Guidance

**Jiaojiao Fan**
Georgia Tech
sbyebss@gmail.com

**Haotian Xue**
Georgia Tech
htxue.ai@gatech.edu

**Qinsheng Zhang**
NVIDIA
qsh.zh27@gmail.com

**Yongxin Chen**
Georgia Tech
yongchen@gatech.edu

## Abstract

There is a rapidly growing interest in controlling consistency across multiple generated images using diffusion models. Among various methods, recent works have found that simply manipulating attention modules by concatenating features from multiple reference images provides an efficient approach to enhancing consistency without fine-tuning. Despite its popularity and success, few studies have elucidated the underlying mechanisms that contribute to its effectiveness. In this work, we reveal that the popular approach is a linear interpolation of image self-attention and cross-attention between synthesized content and reference features, with a constant rank-1 coefficient. Motivated by this observation, we find that a rank-1 coefficient is not necessary and simplifies the controllable generation mechanism. The resulting algorithm, which we coin as `RefDrop`, allows users to control the influence of reference context in a direct and precise manner. Besides further enhancing consistency in single-subject image generation, our method also enables more interesting applications, such as the consistent generation of multiple subjects, suppressing specific features to encourage more diverse content, and high-quality personalized video generation by boosting temporal consistency. Even compared with state-of-the-art image-prompt-based generators, such as IP-Adapter, `RefDrop` is competitive in terms of controllability and quality while avoiding the need to train a separate image encoder for feature injection from reference images, making it a versatile plug-and-play solution for any image or video diffusion model. Our project webpage is https://sbyebss.github.io/refdrop/.

## 1 Introduction

Large-scale diffusion models have demonstrated remarkable capabilities in aiding content creation for artists [43, 7, 3]. Numerous text-to-image models are expediting content production in various domains, including advertising and art studios. Similarly, video generation models have shown significant advancements recently [17, 9, 18, 24, 52, 6, 4, 23]. However, enhancing these models to better support artistic creativity requires improved controllability, particularly in content consistency. This paper explores consistency from two perspectives: 1) controlling subject consistency across multiple images, and 2) maintaining subject consistency across multiple frames within a video.

We name a few tasks where the controllable consistency is crucial in AI content generation. In image generation for storytelling [45, 39, 41, 27] or advertising, content creators often strive to produce consistent characters, a task that proves challenging with foundational generative models [55]. Personalization approaches based on fine-tuning [50] require a minimum of 5 to 10 images to achieve

38th Conference on Neural Information Processing Systems (NeurIPS 2024).

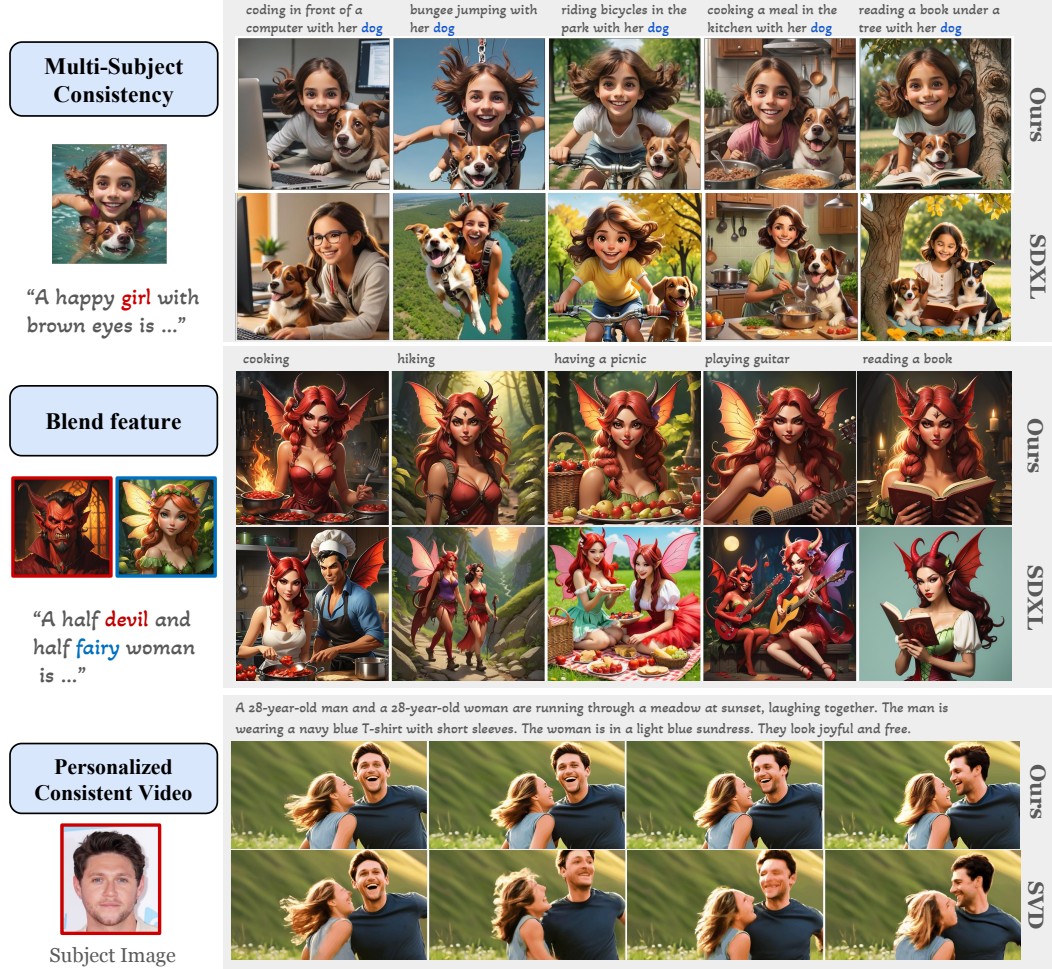

Figure 1: RefDrop achieves controllable consistency in visual content synthesis for free. RefDrop ex-ihibits great flexibility in (Upper) multi-subject consistency generation given one reference image, (Middle) blending different characters from multiple images seamlessly, (Buttom) enhancing temporal consistency for personalized video generation. RefDrop is short for "reference drop". We named our method RefDrop to metaphorically represent the process by which a drop of colored water influences a larger body of clear water.

satisfactory quality, and encoder-based methods [61, 59, 40] demand weeks of training with millions of images for a single diffusion model and lack transferability to other foundational models. On the other hand, diverse image generation is less addressed but persistently challenging. In this scenario, it is desired to *decrease* the consistency among image generations. For example, artists can sometimes seek to enhance diversity and avoid clichés, such as the stereotypical depiction of Barbie girls with curly blonde hair. For video generation, another challenging task is maintaining temporal consistency in video generation, yet most existing solutions are confined to video editing tasks [31], demanding high-quality input videos.

These emerging tasks motivate us to develop RefDrop, a **training-free**, **plug-and-play** method designed to provide flexible control over the consistency in image and video generation. Specifically, we modify the self-attention mechanism in the diffusion model UNet [49] architecture and introduce a coefficient to modulate the influence of a reference image on the generation process. Our contributions are outlined as follows:

1. We conduct a detailed analysis of popular consistency generation methods based on concatenated attention, revealing that their consistency is actually contributed by extra guidance applied implicitly.
2. Inspired by this finding, we propose Reference Feature Guidance (RFG), a natural extension that

explicitly controls the guidance from reference context in a precise and direct manner. Building upon RFG, we introduce RefDrop, a flexible and efficient approach to controlling consistency without the need for network fine-tuning or optimization. 3. Besides improvements in character consistency using a single reference image, RefDrop enables more creative applications with controllable consistency, including (i) seamless integration of distinct features into a single cohesive image (ii) suppressing specific features by negatively decreasing the consistency influenced by the reference context, thereby enhancing diversity in layout, accessories, and image style; (iii) high-quality personalized video generation by boosting temporal consistency, and minimizing facial distortions. 4. We conduct comprehensive experiments and demonstrate that RefDrop achieves a good balance between flexibility and effectiveness while being lightweight compared to existing works.

## 2   Related work

Among the works most similar to ours are IP-Adapter [67] and concatenated attention [62]. Our approach is closely related to IP-Adapter, as both methods utilize the sum of two decoupled attention outputs. However, while IP-Adapter modifies cross-attention and requires separate training of an image encoder to embed the reference image, we integrate the reference image directly into the self-attention layer without needing additional training. Furthermore, our reference images are *generated* by the same model, in contrast to IP-Adapter's reliance on externally sourced image. Both techniques permit the use of negative or positive coefficients for the reference image, but IP-Adapter may compromise text alignment [55] due to its reference image being intertwined with the text prompt during cross-attention. Additionally, the IP-Adapter requires separate training for different versions of the diffusion model, such as SD2.1 and SDXL. In contrast, RefDrop is a simple plug-and-play.

Concatenated attention, first introduced in video generation literature by Wu et al. [62] as spatio-temporal attention, injects temporal information into a T2I model. It has since been widely adopted for feature injection across various applications [37, 8, 25, 55] in content generation and video editing. This concept has evolved into Cross-Frame Attention [29], another prevalent technique used to inflate T2I models [69] for video generation. We will demonstrate later that our framework can replicate these two types of attention as special cases.

A concurrent work by Avrahami et al. [2] introduces a method called soft blending, which is quite similar to our RFG (5), but applies it to a different application: object dragging.

**Consistency in Image Generation**   ConsiStory [55] and StoryDiffusion [71] are closely related to our work. They are training-free methods that employs concatenated attention to enhance consistency in generation. Our RFG framework is *orthogonal* to the techniques other than concatenated attention in those works, such as subject masking and attention dropout. Avrahami et al. [1] explores a fine-tuning-based method aimed at recovering tightly clustered images. Other approaches, such as those by [27, 12, 35], predominantly utilize a personalization process [50, 14, 32, 54] requiring multiple input images for training. Finally, several encoder-based methods [61, 59, 33, 51, 64, 30, 34] do not require additional training for new subjects. However, these methods necessitate days or weeks of initial training for the encoder and face limitations in adaptability to different versions of foundational generative models.

**Temporal-consistency in video generation**   Concatenated attention [62, 47] and Cross-Frame Attention [29, 10] are popular techniques used to inflate T2I models for video generation. Wu et al. [63], Ren et al. [47] mitigate video flickering by applying a low-pass filter to noisy latent images, effectively removing disruptive high-frequency content. Many other methods are tailored for video editing tasks, and they either extract features from high-quality input videos to enhance the current generation [31, 70, 66, 65] or use them as references during editing [16, 11]. RefDrop improves temporal consistency directly within video generation, obviating the need for an input video.

## 3   Method

We first introduce how existing works achieve consistency generation by leveraging the concatenation of reference features in the self-attention block. Then we reformulate the concatenation as a linear interpolation of self-attention on synthesized content and cross-attention between generated and reference content with a constant rank-1 coefficient. We highlight that this specific coefficient is not a

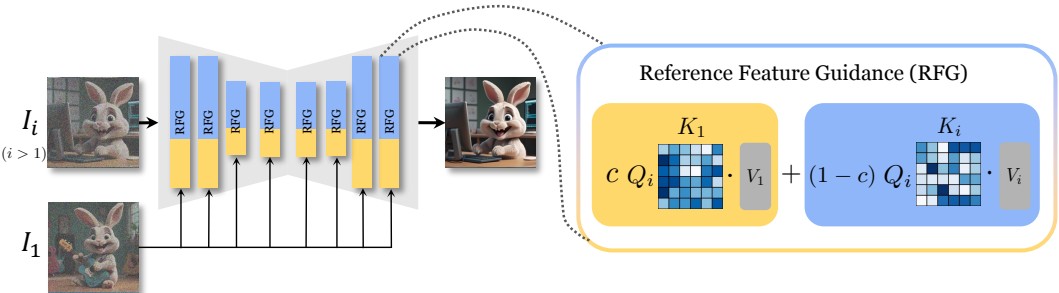

Figure 2: During each diffusion denoising step, we facilitate the injection of features from a *generated* reference image $I_1$ into the generation process of other images through RFG. The RFG layer produces a linear combination of the attention outputs from both the standard and referenced routes. A negative coefficient $c$ encourages divergence of $I_i$ from $I_1$, while a positive coefficient fosters consistency.

necessity, while linear interpolation is critical to minimizing the training-inference gap. Building upon these observations, we propose Reference Feature Guidance (RFG), an extension of concatenation attention that allows for flexible feature interpolation and extrapolation in attention modules. Based on RFG, we introduce RefDrop, a versatile method for controllable consistency generation across various applications.

## 3.1 Background

Self-attention in diffusion model networks operates by applying the attention mechanism [57] on synthesized latent features. A self-attention layer processes latent representations $X$ by passing them through linear projection layers to produce queries $Q = XW_Q$, keys $K = XW_K$, and values $V = XW_V$, which then undergo the attention operation as follows:

$$X' = \text{Attention}(Q, K, V) = \text{Softmax}\left(\frac{QK^\top}{\sqrt{d}}\right) V, \tag{1}$$

where $X'$ is the output of self-attention operation, and $d$ is the feature dimension of projection matrices $W_Q, W_K, W_V$. Previous consistency generation [55, 71] is based on concatenated attention via a simple batch image generation, where the first sample in the batch serves as a reference for the $i$-th sample generation. We denote the latent feature for $i$-th sample as $X_i$. Instead of solely depending on its own content, concatenated attention suggests

$$X'_{\text{CAT}} = \text{Attention}\left(Q_i, [K_1; K_i], [V_1; V_i]\right), \tag{2}$$

where $Q_i = X_iW_Q$, $K_i = X_iW_K$, and $V_i = X_iW_V$.

## 3.2 Reference feature guidance

To illustrate why concatenated attention can help boost consistency between generated samples with reference samples, we can reformulate eq. (2) as the following (Proof in appendix C)

$$X'_{\text{CAT}} = C \odot \text{Attention}\left(Q_i, K_1, V_1\right) + (\mathbf{1} - C) \odot \text{Attention}\left(Q_i, K_i, V_i\right) \tag{3}$$

where $C$ is a rank-1 matrix of the same size as the attention output, $\odot$ is the point-wise multiplication and $\mathbf{1}$ is an all-ones matrix.

Equation (3) depicts that the concatenated attention is a linear interpolation between the output $X'$ without concatenated attention in eq. (1) and cross-attention between the $i$-th image $X_i$ and the reference image $X_1$, while the coefficient matrix $C$ is determined by the synthesized content $X_i$ and the reference content $X_1$. Before we further improve concatenated attention, we first discuss two related questions for eq. (3). **Is linear interpolation a necessity?** It may be tempting to highlight the role of the second cross-attention term naively while keeping the weights for the first term unchanged, such as $\text{Attention}(Q_i, K_1, V_1) + \text{Attention}(Q_i, K_i, V_i)$. However, we find that naively breaking the linear interpolation disrupts image generation. In fact, we can interpret concatenated attention in eq. (3) as applying extra guidance on the original self-attention output

$$X'_{\text{CAT}} = \text{Attention}\left(Q_i, K_i, V_i\right) + C \odot \left(\text{Attention}\left(Q_i, K_1, V_1\right) - \text{Attention}\left(Q_i, K_i, V_i\right)\right) \tag{4}$$

which resembles the form of guidance used in diffusion literature [53, 22], such as classifier-free guidance [21]. Notably, the linear interpolation helps keep the attention output $X'_{\mathtt{CAT}}$ norm close to self attention output $X'$; otherwise, arbitrary weights would pose a training and inference discrepancy and degrade the generation quality. However, different from various guidance methods used in the diffusion literature, the guidance weights in eq. (4) are constants determined by latent features $X_i$ and reference context $X_1$ and have no user control. Therefore we question **Is constant $C$ matrix coefficient is a necessity?** As an attempt to bypass the rigid form of concatenated attention, we propose a simple and flexible approach named *Reference feature guidance* (RFG) (see Fig. 2)

$$X'_{\mathtt{RFG}} = c \cdot \text{Attention}\,(Q_i, K_1, V_1) + (1 - c) \cdot \text{Attention}\,(Q_i, K_i, V_i), \tag{5}$$

where $c$ is a scalar coefficient that controls the strength of the reference image influence.

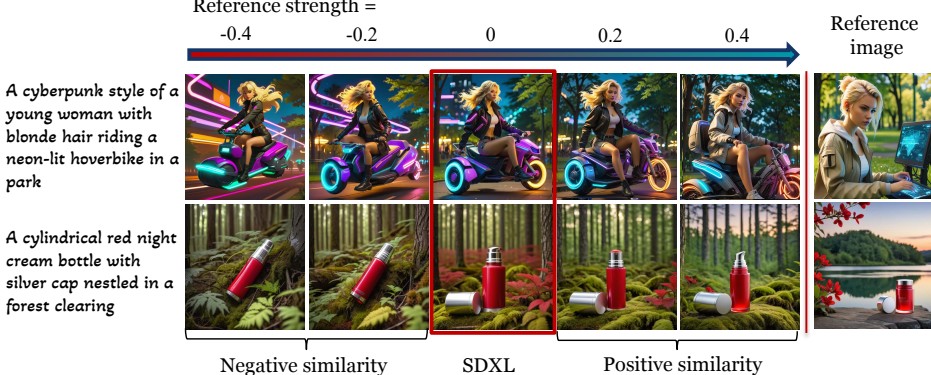

Figure 3: We allow flexible control over the reference effect through a reference strength coefficient.

While most previous methods for consistent generation, including feature combination [30, 71] and injection [55], employ concatenated attention (2), our RFG offers several advantages. First, it grants users greater control over the extent of influence from the reference image, as illustrated in Fig. 3. Second, this flexibility proves especially beneficial in a novel application: blending features from multiple reference images. Our method allows users to selectively determine the influence of each reference image. We have observed that the most harmonious blending often results from varying the strength of each reference, rather than maintaining equal strength across all images. Third, by enabling negative coefficients, we find that our method can simulate a concept suppression effect, meaning it generates images that are dissimilar to a reference image. Moreover, it allows for the injection of reference image features into video generation slightly to reduce flickering, while the concatenated attention keeps the video completely static. Finally, our approach is versatile on network architecture as it applies not only to UNet-based models but also to transformer-based diffusion models, such as FLUX model (see appendix D.1).

Therefore, we introduce RefDrop, a training-free approach to flexibly control consistency generation, which replaces the self-attention blocks in the diffusion model with RFG. For Video Diffusion Models (VDM) [5, 15, 19], we modify *every spatial* self-attention layers to bolster temporal consistency.

## 4 Experiments

We conduct experiments to show that RefDrop can help control consistency in two important tasks: image generation and video generation.

### 4.1 Controllable consistency in image generation

We use a fine-tuned SDXL of higher quality, ProtoVision-XL, as the base model for our experiments. For simplicity, we will refer to it as

Table 1: Comparison of Controllable Consistent Image Generation Methods. 'Training-free' indicates no encoder training or diffusion model fine-tuning is needed. 'Single ref.' means the method can operate with only one reference image.

| Name | Training free | Concept suppression | Single ref. |
|---|---|---|---|
| IP-Adapter [67] | ✗ | ✓ | ✓ |
| Consistory [55] | ✓ | ✗ | ✓ |
| Chosen one [1] | ✗ | ✗ | ✗ |
| ELITE [61] | ✗ | ✗ | ✓ |
| BLIPD [33] | ✗ | ✗ | ✓ |
| Ours | ✓ | ✓ | ✓ |

SDXL hereafter. We have replaced all the self-attention layers in SDXL with RFG, using the first sample in the batch as the reference image.

**Evaluation baselines**  In this section, we compare RefDrop with several baseline approaches: (1) SDXL [43] without any modifications to its architecture; (2) Ref-ControlNet [1]; (3) encoder-based methods, such as IP-Adapter [67] and BLIPD [33]. For encoder-based methods, we initially generate a reference image using SDXL and then utilize this image as input. Additionally, we present a comparison of several other methods in Table 1.

### 4.1.1 Consistent image generation

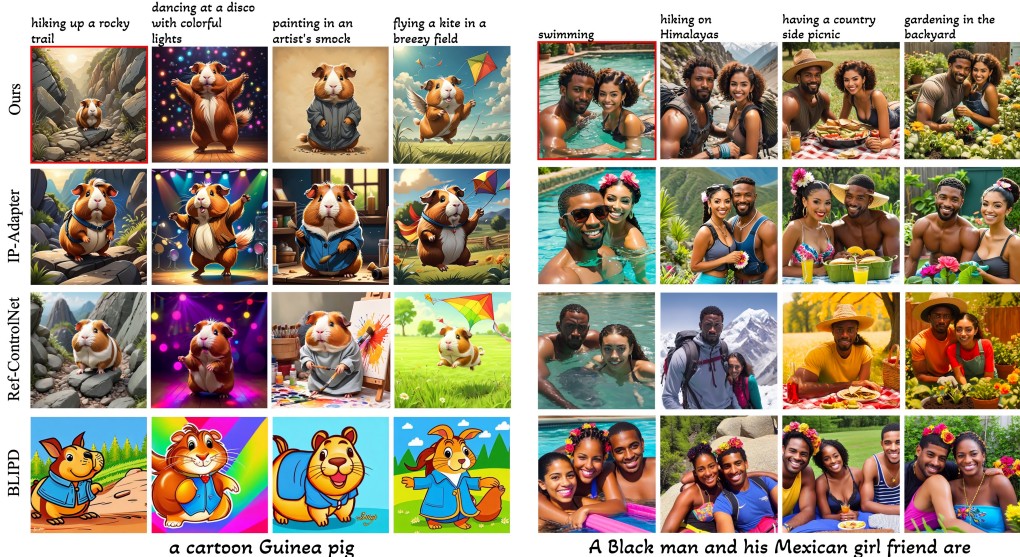

Figure 4: The reference image for all methods is framed in red. Our method tends to produce more consistent hairstyles, and facial features compared to IP-Adapter, Ref-ControlNet and BLIPD, and our generation has diverse spatial layout. The visual quality of BLIPD is not comparable, as it utilizes SD1.5 [48] as its base model.

For this task, we use $c \in [0.3, 0.4]$ for our method. However, applying RFG to all the self-attention blocks can lead to the leakage of spatial layout and background from the reference image, causing the generated objects to have quite similar poses and backgrounds. To address these issues, we introduce two techniques: excluding the first upsampling block and applying the subject mask.

**Excluding the first upsampling block** SDXL UNet consists of 4 downsampling blocks, 1 middle block, and 6 upsampling blocks. Through an ablation study, we found that the first upsampling block predominantly influences the spatial layout. As shown in Fig. 5, excluding this block from the modified attention blocks allows for recovering diverse object poses. To the best of our knowledge, this is the first work to use this method for mitigating spatial layout leakage in consistent image generation. Consistory [55] also proposes two techniques to enhance layout diversity: using vanilla query features and self-attention dropout. In comparison, our approach is simpler and more straightforward.

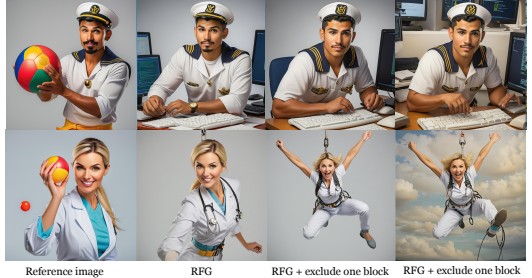

Figure 5: Excluding one block from applying RFG solves the spatial layout leakage issue. Adding subject mask solves the background leakage issue.

**Applying the subject mask** We use Grounded SAM [46] to extract the object mask from the generated reference image by prompting the object name, such as "Guinea pig" or "human." The

[1] https://github.com/Mikubill/sd-webui-controlnet/discussions/1236

mask is then downsampled to match the latent feature resolution of the SDXL UNet. The resulting masked RFG is defined as follows

$$X'_{\texttt{RFG}} = cM \odot \text{Attention}\,(Q_i, K_1, V_1) + (\mathbf{1} - cM) \odot \text{Attention}\,(Q_i, K_i, V_i),\qquad(6)$$

where the mask $M$ ensures that guidance is restricted to the masked area. Note that, our masked RFG (6) does not modify the attention operation itself but only adjusts the coefficients, making it memory efficient and straightforward to implement.

We show qualitative results in Fig. 4. IP-Adapter establishes a strong baseline, especially on single subject consistent generation. However, it suffers from similar spatial layout, and requires additional computational resources and data for training the image encoder compared to our approach. While Reference-only ControlNet performs well on simple subjects, such as cartoon characters, it struggles to generate humans. It is likely to produces humans with distorted eyes and bodies. BLIPD underperforms in terms of both visual quality and consistency relative to RefDrop. For **multi-subject** consistent generation, we find RefDrop can straightforwardly work for semantically different objects even without using separate subject masks. This observation aligns with ConsiStory [55]. Further comparative results are available in Figs. 13 and 16.

### 4.1.2 Blend features from multiple images

RefDrop also supports the use of multiple reference images. In our implementation, we designate the first $N$ images in a batch as reference images. Features from these reference images are then incorporated into the subsequent images within the same batch through every self-attention layer. Formally, the extended RFG with multiple references is defined as

$$X'_{\texttt{RFG}} = \sum_{j=1}^{N} c_j \cdot \text{Attention}(Q_i, K_j, V_j) +$$
$$(1 - \sum_{j=1}^{N} c_j) \cdot \text{Attention}(Q_i, K_i, V_i),\qquad(7)$$

for certain $i > N$. Here, the attention mechanism ensures that the $i$-th image in the batch receives features from the first $1 \sim N$ reference images. In practice, we use $c_j \in [0.2, 0.4]$ for any $j = 1, \ldots, N$ in our method. We demonstrate the capability of RefDrop to seamlessly

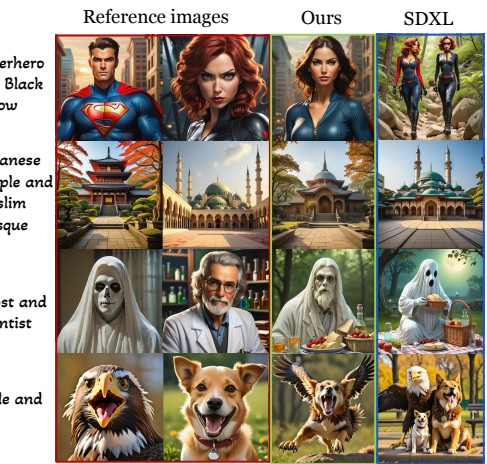

Figure 6: **Multiple Reference Images:** The reference images are highlighted with a red frame, and the third image in each set is the resultant blended image. RefDrop effectively assimilates features from the distinct reference images into a single and cohesive entity, demonstrating robust feature integration capability.

blend distinct semantic features from two reference objects into a new object in Figs. 6, 18 and 19. This task proves challenging when relying solely on prompt engineering. We attempted to achieve this task with SDXL using text prompts. For instance, if we aim to merge two objects, $\alpha$ and $\beta$, we might use prompts like "an $\alpha$-like $\beta$" or "a $\beta$ in the style of $\alpha$." However, with such text prompts, SDXL either ignores the similarity with one of the reference images or frequently produces multiple objects instead of a single and cohesive entity. In Fig. 20, we show that RefDrop can blend *three* distinct subjects: a dwarf, Black Widow, and Winnie the Pooh, encompassing a range of mythological being, human, and animal.

### 4.1.3 Diverse image generation

Our method offers substantial flexibility in parameter tuning, enabling diverse image generation by setting the coefficient $c$ to a negative value. This feature is particularly valuable in addressing overfitting issues in image generation. For instance, when using SDXL to generate Middle Eastern faces, the output frequently includes similar headscarves, faces and outfits, as illustrated on the left side of Fig. 8.

Figure 7: Negative reference images for examples in Fig. 8.

In this task, we use $c = -0.3$ for our method. We present a qualitative comparison in Fig. 8, with negative reference images displayed in Fig. 7. Upon comparing our method with IP-Adapter, we

note that IP-Adapter may not adhere as closely to the text prompt. We attribute this to IP-Adapter's modification of cross-attention, which can impact text alignment. In contrast, our method focuses on modifying self-attention, thereby preserving the integrity of cross-attention and ensuring more accurate text alignment. We show additional quantitative results in Fig. 15.

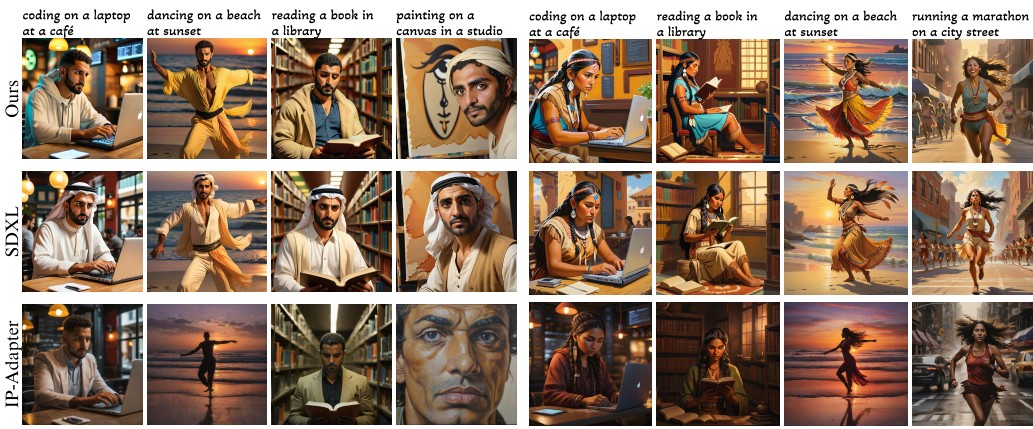

Figure 8: Diverse image generation: Our method enhances diversity in outfits, hairstyles, and facial features, all while ensuring accurate text alignment. For example, while SDXL frequently generates headscarves in the first scenario and beige-colored clothes in the second, `RefDrop` can vary the presence of headscarves in the left example and produce clothing in different colors in the right example. Conversely, although IP-Adapter can create even more diverse images, it often fails to adhere to the style and human activity instructions in the text prompts. Additionally, it often produces overly small persons that lack detail.

## 4.2 Improving temporal-consistency in video generation

Not only can we apply `RFG` to T2I generation, but it also effectively stabilizes video generation, where flickering commonly degrades quality. This section shows that using the first generated frame as a reference can greatly improve video generation. By injecting its features into the spatial self-attention layers of subsequent frames with a reference strength of $c = 0.2$, we significantly stabilize these frames and enhance the temporal consistency of VDM.

We employ `SVD-img2vid-xt-1-1` as our I2V base model. Technically, our approach is compatible with any VDM, but we choose SVD as it is the best open-source model available. Although this model usually produces consistent videos from visually perfect images, we have noted that minor, often imperceptible flaws in the input images can significantly degrade the quality of the generated videos. Our method effectively stabilizes video quality in these scenarios.

### 4.2.1 Temporal-consistent video generation

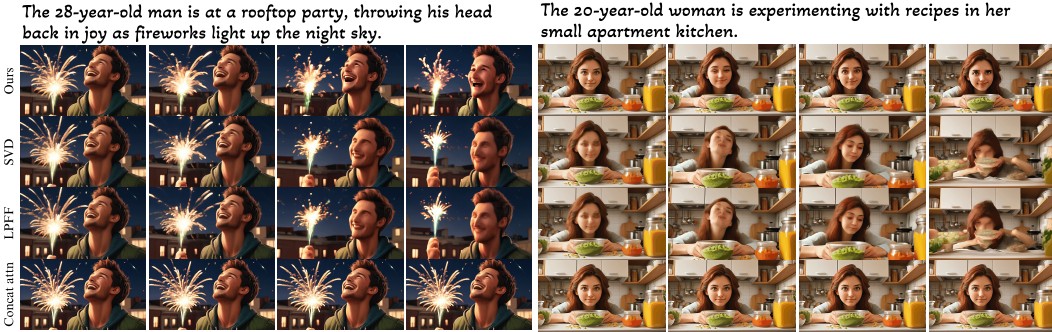

Figure 9: Comparison of training free techniques to improve temporal consistency in video generation.

In this part, we use the SDXL model to generate an image from a prompt and then pass this image to the SVD model. For **evaluation baselines**, we compare `RefDrop` with several training-free methods: unmodified SVD, Cross-Frame Attention [29], Concatenated Attention [62], and Temporal Low Pass Frequency Filter (LPFF) [69]. Temporal LPFF is arguably superior to Spatial-Temporal LPFF by Wu et al. [63], which shows Spatial-Temporal LPFF can result in blurry frames. We evaluate the Temporal LPFF using a fast sampling method that avoids the computationally intensive process of iteratively performing backward and forward diffusion at each denoising step.

The visualization results are displayed in Fig. 9. We observe that both Cross-Frame Attention and Concatenated Attention result in completely static videos, whereas LPFF shows minimal improvement. Our method proves to be the most effective in preventing flickering while preserving motion.

### 4.2.2 Stabilizing personalized video generation

Finally, we explore the application of `RefDrop` to personalized video generation. Inspired by Ku et al. [31], starting with an image of a person, we use InstantID [59] to generate a personalized initial frame. This frame is then fed into SVD to create a short video. However, we observe that using the output from InstantID for SVD generation leads to a significantly higher failure rate compared to using the initial frame generated by SDXL. We attribute this increased failure rate to InstantID's propensity for producing images with more flaws, such as overly saturated colors, and distorted limbs, highlighting the potential demand for `RefDrop` in this task.

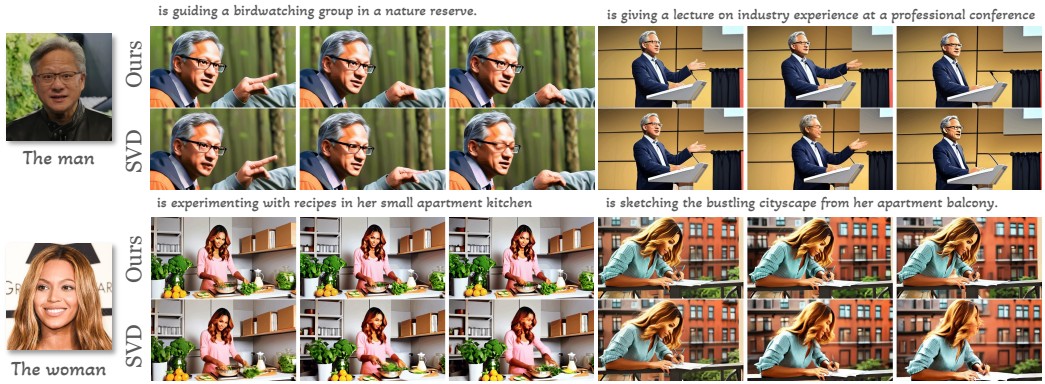

Figure 10: By injecting the features of the first frame into the generation of subsequent frames, `RefDrop` reduces flickering and facial distortions. The additional videos can be viewed here.

Several other methods are available for personalized video generation, as described in works [60, 28, 38, 24]. Our method, which is designed to enhance temporal consistency, can be integrated with some of these existing approaches. For example, in the case of Magic-Me [38], our attention mechanism `RFG` can be incorporated into their AnimateDiff [18] backbone. For the evaluation in this section, we primarily focus on comparisons with naive SVD generation, as it directly relates to our goal of enhancing temporal consistency.

We present such comparison between our `RefDrop` enhanced generation to the naive SVD generation in Fig. 10. `RefDrop` effectively preserves identity during video generation, offering improvements similar to those achieved by increasing the CFG. However, unlike increasing CFG, which often results in over-saturation of videos, our approach does not produce such artifacts. We present additional automatic metrics in Table 3 to show that `RefDrop` can enhance the quality of the generated videos.

## 5 Human evaluation

We conducted a human evaluation study using Google Forms. Our survey is structured into three distinct categories: 1) Consistent Image Generation, 2) Diverse Image Generation, and 3) Personalized Video Generation. Initially, we utilized ChatGPT to generate text prompts, then processed approximately 100 small tasks per category using both baseline methods and our approach. From these, we randomly selected 10 sets for evaluation. In the first two categories, participants assessed the

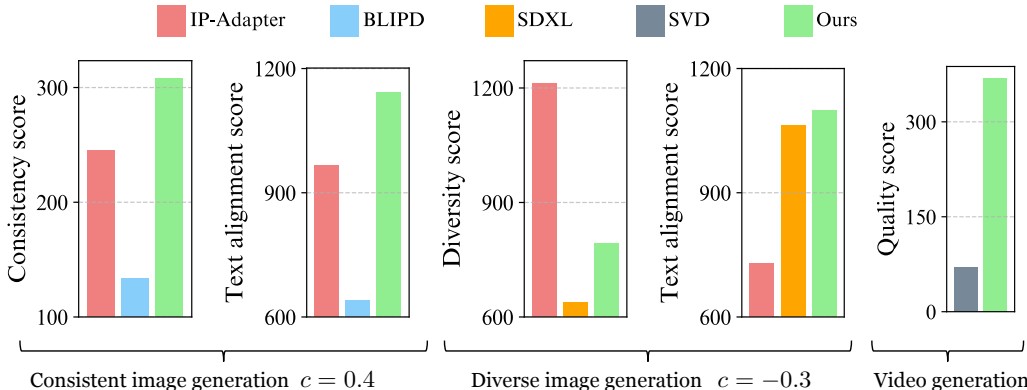

Figure 11: A higher score indicates a superior result. In the category of consistent image generation, participants showed a preference for `RefDrop`, with IP-Adapter ranking slightly behind. In diverse image generation, while IP-Adapter was favored for its variety, it significantly compromised text alignment. Conversely, `RefDrop` maintained a good balance, achieving diversity while preserving text alignment. In personalized video generation, users clearly preferred our approach, demonstrating substantial improvements over the SVD results.

consistency and diversity of the images, as well as text alignment. For the third category, participants were asked to select the video with better quality. The vertical axes in Fig. 11 mean the aggregated scores from all participants. We collected responses from 44 distinct users in total. More details appear in appendix G.

## 6    Conclusion

In this study, we propose a method that effectively uses one or multiple generated images to guide the generation of other images or video frames. Through extensive experiments, our method has proven useful for flexible consistency control in image generation and has improved temporal consistency in video generation. In particular, we show applications in consistent and diverse image generation, feature blending from multiple images, and enhancement of video temporal consistency. Moreover, our approach is versatile on network architecture as it applies not only to UNet-based models but also to transformer-based diffusion models like DiT [42].

Looking ahead, several promising avenues for further research emerge from this study. Firstly, our experiments have not yet explored the use of attention masks; investigating their potential for precise control in image generation presents a compelling opportunity for future work. Another exciting prospect involves enhancing our method to accept clean reference images as input, similar to the IP-Adapter and other image personalization techniques. Achieving this capability would represent a significant advancement, particularly if coupled with an optimal image inversion method.

## Acknowledgments and Disclosure of Funding

We thank Yuval Atzmon and the anonymous reviewers for their valuable feedback on this paper. Jiaojiao Fan, Haotian Xue, and Yongxin Chen are supported in part by grants NSF CAREER ECCS-1942523, and NSF FRR-2409016.

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

## A  Limitation

- For consistent image generation, our model sometimes struggles to accurately replicate specific objects like lotion bottles, often hallucinating instead. It also occasionally fails to replicate the exact hairstyle and outfit from the reference person or animal.

- For diverse image generation, our method currently cannot precisely control aspects of diversity, such as generating diverse subjects but not styles, or vice versa.

- For blending multiple images, the coefficient needs careful tuning to achieve a good balance among multiple reference images, and it heavily depends on the specific case.

- For improving video temporal consistency, although our method successfully generates videos with temporal consistency, the generated videos contain less motion. A straightforward way to alleviate this issue is to adjust the RFG coefficient depending on the case. To fully address this issue, we believe introducing additional training-based modules, for example, a motion guidance module proposed in StoryDiffusion [71], could provide stronger motion guidance. Alternatively, we could add motion frames as conditional guidance, as done in EMO [56].

- Finally, when applying our method to models beyond SDXL or SVD, we find that the coefficient parameter $c$ may require additional tuning. Additionally, for the technique we proposed to address spatial layout leakage, an ablation study is necessary to identify which layer primarily governs spatial layout generation, as this layer may vary across different models.

## B  Broader impacts

The broader impacts of advancements in consistent character generation and personalized video generation extend across multiple domains, notably enhancing both creative and technological landscapes. In the media and entertainment industries, for instance, these methods can revolutionize character design, fostering more reliable representations. For media and advertising, it can enhance the design of key frames in advertisements or movie videos, allowing companies to create more visually compelling content. Individual artists may leverage this technique to blend multiple images, potentially sparking new creative inspirations. RefDrop could be used to improve temporal consistency in short videos, which may be particularly valuable for social media platforms seeking to enhance AI-generated video content. However, there is also a potential risk associated with our method, as it could be used to create fake profiles, highlighting the need for careful consideration of its applications.

## C  Relationship to Concatenated attention

### C.1  Mathematical equivalence

We firstly point out that our `RFG` framework can recover the concatenated attention (2) by replacing the scalar coefficient in (5) to be a rank-1 matrix. Then we delve into details. We begin by noting the dimensions of the attention maps:

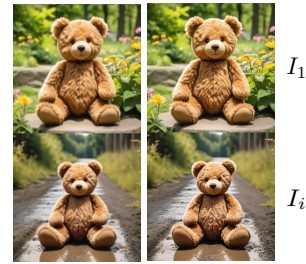

$I_1$

$I_i$

$$\text{Softmax}\left(\frac{Q_i K_1^\top}{\sqrt{d}}\right) \in \mathbb{R}^{L \times L}, \tag{8}$$

$$\text{Softmax}\left(\frac{Q_i K_i^\top}{\sqrt{d}}\right) \in \mathbb{R}^{L \times L}, \tag{9}$$

Eq. (11)-(13)   Concat attn (2)

Figure 12: Concatenated attention is our special case.

$$\text{Softmax}\left(\frac{Q_i [K_1; K_i]^\top}{\sqrt{d}}\right) \in \mathbb{R}^{L \times 2L}, \tag{10}$$

where $L$ is the sequence length of the input hidden feature $X$. We further define $\mathbf{1}_d$ as an all-ones vector of dimension $d$, and $\mathbf{1}$ as an all-ones matrix, sized appropriately to ensure the validity of the operations it is involved in. To recover concatenated attention (2), we extend the scalar $c$ from (5) to a rank-1 weight matrix:

$$C = \mathbf{c} \otimes \mathbf{1}_{d_v}, \tag{11}$$

where all columns in $C \in \mathbb{R}^{L \times d_v}$ are identical, represented by the vector $\mathbf{c} \in \mathbb{R}^{d_v}$, and $d_v$ is the feature dimension of the value $V$. We then transform the scalar dot product into a matrix element-wise product $\odot$, allowing RFG to be expressed with this matrix coefficient as:

$$X'_{\text{RFG}} = C \odot \left( \text{Softmax}\left( \frac{Q_i K_1^\top}{\sqrt{d}} \right) V_1 \right) + (\mathbf{1} - C) \odot \left( \text{Softmax}\left( \frac{Q_i K_i^\top}{\sqrt{d}} \right) V_i \right). \quad (12)$$

Denote $./$ and $\exp$ as the element-wise division and exponential operation respectively. By setting

$$\mathbf{c} = \left( \exp\left( \frac{Q_i K_1^\top}{\sqrt{d}} \right) \mathbf{1}_L \right) ./ \left( \exp\left( \frac{Q_i [K_1; K_i]^\top}{\sqrt{d}} \right) \mathbf{1}_{2L} \right), \quad (13)$$

$X'_{\text{RFG}}$ can recover the concatenated attention [62]

$$X'_{\text{CAT}} = \text{Softmax}\left( \frac{Q_i [K_1; K_i]^\top}{\sqrt{d}} \right) [V_1; V_i] = \text{Softmax}\left( \frac{[Q_i K_1^\top; Q_i K_i^\top]}{\sqrt{d}} \right) [V_1; V_i]. \quad (14)$$

The reason for this to hold is simply the normalizing effect of softmax. The softmax operation would normalize each row in the attention map $\frac{Q_i [K_1; K_i]^\top}{\sqrt{d}}$ independently, thus the weight matrix to recover the concat attention is a rank-1 matrix with different rows.

We want to note that the rank-1 matrix used in the concatenated attention-based method is not explicitly shown because they use Equation (2). One of our contributions is demonstrating that Equation (2) can be equivalently reformulated as Equation (3), where the rank-1 matrix appears as the coefficient. We further simplify Equation (3) by changing the rank-1 matrix to a scalar. We emphasize that in the concatenated attention-based method, the rank-1 matrix is not manually adjustable or defined by the user; it is intrinsically determined by the reference image feature and generated image feature.

We also note that if we only use a scalar as the coefficient, we cannot exactly replicate the concatenated attention method. However, we empirically find that using a coefficient of 0.3~0.4 can produce a similar effect to the concatenated attention method.

Finally, we can also recover the Cross-Frame attention [29] by setting the coefficient $c = 1$.

## C.2 Visual comparison

In Fig. 13, we present a visual comparison with concatenated attention for consistent image generation task, using the same random seed and without additional techniques such as excluding one attention block (as proposed by us) and self-attention dropout from Consistory [55]. In practice, we find that RFG with a coefficient of 0.3~0.4 produces results quite similar to the concatenated attention method used in Tewel et al. [55], Zhou et al. [71].

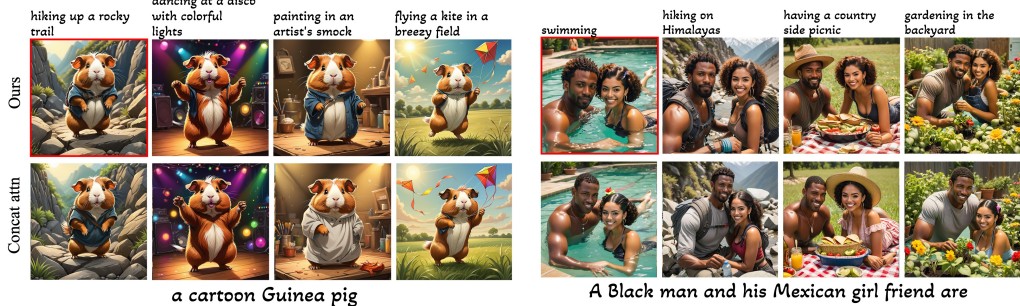

Figure 13: Comparison with Concatenate attention for the consistent image generation task. Both of ours and Concatenated attention methods do not apply subject mask and are applied to all attention blocks here for a fair comparison.

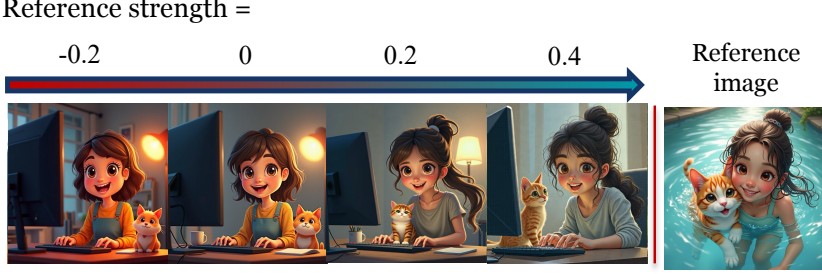

Figure 14: Applying Reference Feature Guidance on the FLUX-dev model. Our method effectively applies reference guidance to the generation.

# D  Additional results

## D.1  Applying RFG on transformer-based architecture

We apply Reference Feature Guidance on the most recent capable open-source T2I model, FLUX-dev, developed by Black Forest Lab. We show the effect of Reference Feature Guidance on the FLUX-dev model in Fig. 14. Since FLUX is a transformer-based diffusion model, it does not have separate cross-attention and self-attention layers. we have to do some modification to adapt to their model. Every attention block in FLUX-dev is:

$$X' = \text{Attention}([Q_{\text{img}}; Q_{\text{txt}}], [K_{\text{img}}; K_{\text{txt}}], [V_{\text{img}}; V_{\text{txt}}])$$

where $[\cdot; \cdot]$ denotes the concatenation operation. Similar to (4), we modify the attention blocks in FLUX to accept additional guidance from the reference image feature, assumed to be the first image in the batch:

$$X'_{\text{RFG}} = X' + c \cdot (\text{Attention}(Q_{\text{img } i}, K_{\text{img } 1}, V_{\text{img } 1}) - \text{Attention}(Q_{\text{img } i}, K_{\text{img } i}, V_{\text{img } i})) \quad (15)$$

We add two additional terms multiplied by a guidance scale $c$ on top of the original attention output $X'$. And those additional terms only depend on the image features, ensuring that we do not interfere with the text features. Fig. 14 demonstrates that our method effectively applies reference guidance to the generation.

## D.2  Quantitative results

**Importance of the first upsampling block for SDXL UNet**  In Sec. 4.1.1, we introduced a technique to mitigate the spatial layout leakage from the reference image. In Table 2, we show quantitative result to verify that removing the first upsampling block is more effective than removing other blocks here to remove spatial layout leakage. We conducted 11 groups of experiments, where in each group, we excluded one of the 11 blocks of the UNet from applying RFG. And in each group, we generated 20 sets of consistent objects, with each set containing 5 images. This resulted in a total of $11 \times 20 \times 5 = 1100$ images for metric calculation. We then used DreamSim and LPIPS to measure the distance between the generated images and the reference image, and report the mean and standard deviation. Higher values of these metrics should indicate more diverse poses, as these scores tend to favor similar layouts [55]. From Table 2, we can see that excluding the first upsampling block greatly boosts the spatial layout diversity.

**Text-to-Image generation**  We present quantitative metrics in Figure 15. Using the OpenCLIP model, CLIP-ViT-g-14-laion2B, we measure text-image similarity by averaging CLIP scores [20] across 100 pairs of text prompts and generated images. This measurement is repeated five times using different pairs for each method, and the variability is depicted through error bars. For assessing subject consistency, we utilize DreamSim [13], after processing images to remove backgrounds[2]

---

[2]We use the Tracer-B7 model in
https://github.com/OPHoperHPO/image-background-remove-tool/?tab=readme-ov-file

Table 2: Comparison of DreamSim and LPIPS distances for excluding different blocks. The Up1 block of SDXL UNet shows the highest values for both metrics, indicating its strong impact on spatial layout diversity.

| Excluded Block | DreamSim to reference image | LPIPS to reference image |
|---|---|---|
| Down1 | $0.2283 \pm 0.0102$ | $0.5484 \pm 0.0052$ |
| Down2 | $0.2261 \pm 0.0118$ | $0.5484 \pm 0.0043$ |
| Down3 | $0.2298 \pm 0.0082$ | $0.5475 \pm 0.0058$ |
| Down4 | $0.2332 \pm 0.0095$ | $0.5551 \pm 0.0072$ |
| Mid | $0.2484 \pm 0.0048$ | $0.5673 \pm 0.0070$ |
| Up1 | $\mathbf{0.3077 \pm 0.0092}$ | $\mathbf{0.5880 \pm 0.0047}$ |
| Up2 | $0.2567 \pm 0.0087$ | $0.5603 \pm 0.0077$ |
| Up3 | $0.2440 \pm 0.0088$ | $0.5580 \pm 0.0032$ |
| Up4 | $0.2441 \pm 0.0079$ | $0.5611 \pm 0.0074$ |
| Up5 | $0.2368 \pm 0.0106$ | $0.5540 \pm 0.0055$ |
| Up6 | $0.2455 \pm 0.0103$ | $0.5639 \pm 0.0075$ |

in order to focus analysis on foreground content. In tasks of diverse image generation, we employ LPIPS to gauge image diversity. We calculate the pairwise DreamSim or LPIPS distance between 400 image pairs per method, repeating these measurements with distinct pairs to ensure robust results, and report these findings with error bars. The measures of consistency and diversity are expressed as one minus the calculated DreamSim or LPIPS distances. These results demonstrate that our method is effectively situated on the Pareto-front, aligning with the human evaluations reported in Figure 11.

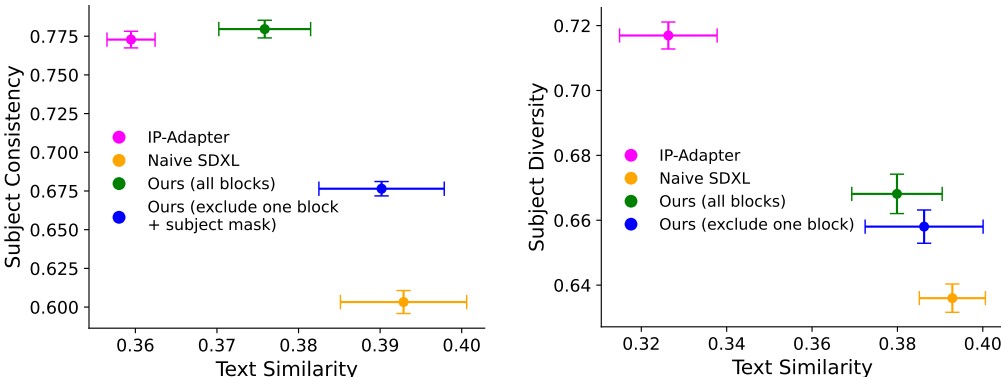

Figure 15: Left: **Consistent Image Generation.** Our method achieves a good balance between text alignment and subject consistency. The techniques of excluding the first upsampling block from being influenced by the reference image (see Sec. 4.1.1) and adding subject masks (see Sec. 4.1.1) help achieve diverse spatial poses and resolve background issues, leading to significant improvements in text alignment. Right: **Diverse Image Generation.** Our approach maintains higher subject diversity with only a slight compromise in text alignment. In contrast, IP-Adapter exhibits the highest subject diversity but suffers from a significant reduction in text alignment. The techniques of excluding the first upsampling block (see Sec. 4.1.1) can also help improve the text alignment here. Error bars represent standard deviation.

**Image-to-Video generation**    We present a comparison of automatic metrics for video generation in Table 3. All metrics are designed by EvalCrafter [36]. Following EvalCrafter, we measure the quality of generated videos from four perspectives: overall quality, text alignment, temporal consistency, and motion quality. Specifically, $VQA_A$ measures the aesthetic score, and $VQA_T$ evaluates common

distortions such as noise and artifacts. CLIP Score quantifies the similarity between input text prompts and generated videos. For temporal consistency, we use CLIP-Temp to measure semantic consistency between frames, and also calculate face consistency, and warping errors. Finally, the flow score calculates the average optical flow across all video frames. We generated 220 personalized videos using 220 distinct prompts for both SVD and `RefDrop`, utilizing images of four individuals shown in Fig. 10. The prompts included both close-up and distant descriptions. The metrics shown in Table 3 are averaged over these 220 videos. The statistics demonstrate that `RefDrop` reduces unnecessary flickering and improves overall quality. Surprisingly, we find that `RefDrop` not only improves the visual quality but also the text alignment.

Table 3: Comparison of automatic metrics between SVD and `RefDrop` on video generation. An ↑ symbol indicates that higher values are better, while a ↓ symbol indicates that lower values are preferable. Our model shows improvements over the SVD base model in overall quality, text alignment, and temporal consistency. The flow score is the only metric where the SVD model scores higher, indicating more motion. However, the SVD model also exhibits greater jittering and flickering, as reflected in its larger warping error. Notably, a static video would register a flow score of zero. This suggests that our generated videos maintain a reasonable level of motion.

| | Overall quality | | Text alignment | Temporal Consistency | | | Motion |
|---|---|---|---|---|---|---|---|
| | $VQA_A$ ↑ | $VQA_T$ ↑ | CLIP score ↑ | CLIP Temp ↑ | Face consis. ↑ | Warping error ↓ | Flow score ↑ |
| Ours | **94.27** | **89.91** | **20.84** | **99.91** | **99.46** | **0.0058** | 2.62 |
| SVD | 93.25 | 86.20 | 20.76 | 99.83 | 99.20 | 0.0077 | **5.80** |

### D.3 Qualitative results

**Consistent and Diverse Image Generation:** We give more visualizations for consistent and diverse image generation in Fig. 16 and Fig. 17. We attached images their original quality in `consistent_generation_remove_up1_mask.pdf` and `diverse_generation.pdf` in the supplementary material.

**Blend multiple images:** We show additional blended images using multiple reference images in Fig. 18, Fig. 19, and Fig. 20. In particular, Fig. 18, Fig. 19 utilize two reference images, and Fig. 20 blends three reference images.

**Personalized Video Comparisons:** We show additional comparison in Fig. 21. Moreover, we offer more than 20 original videos in $1024 \times 576$ resolution, accessible via this anonymous external link. On the linked page, the left column displays the video generations of SVD, while the right column features the enhanced SVD results by `RefDrop`.

## E  Effect of the coefficient $c$

- **Consistent Image Generation:** More challenging tasks typically require larger coefficients to ensure consistency. For example, generating human figures, which are more complex, requires coefficients between $[0.3, 0.4]$. In contrast, simpler subjects like fluffy toys or cartoon characters may only need a coefficient of $0.2$ to achieve consistent generation.
- **Blend multiple images:** We find that the coefficients for each reference image, typically falling within the range of $[0.2, 0.4]$, perform effectively.
- **Diverse Image Generation:** We recommend using a coefficient of $c = -0.3$. Lower strengths can impair visual quality and may introduce artifacts.
- **Video Consistency:** The coefficient for video consistency requires more nuanced control; A coefficient of $0.2$ generally suffices, and a larger coefficient may make the video totally static. This sufficiency is likely due to the temporal attention component in VDM, which tends to amplify the effects introduced through self-attention.

The effect of reference strength on image generation is in Figs. 3 and 14.

# F   Additional implementation details

All experiments were conducted on a single NVIDIA A100 GPU with 80GB of memory. The generation process for a single image using SDXL requires approximately 5 seconds, whereas generating a video using SVD takes about 30 seconds. Additional details on hyper-parameters for both baseline methods and our approach are provided in Table 4.

Table 4: Base model and hyper-parameters.

|  | Base model | CFG | Our reference strength | IP-Adapter scale | TLPFF |
|---|---|---|---|---|---|
| Sec. 4.1.1 | Protovision-XL | 5 | 0.3~0.4 | 0.6 | N/A |
| Sec. 4.1.2 | Protovision-XL | 5 | 0.2~0.4 | N/A | N/A |
| Sec. 4.1.3 | Protovision-XL | 5 | -0.3 | -0.6 | N/A |
| Sec. 4.2.1 | SVD-img2vid-xt-1-1 | 2.5 | 0.2 | N/A | Gaussian filter Stop frequency = 0.5 |
| Sec. 4.2.2 | SVD-img2vid-xt-1-1 | 2.5 | 0.2 | N/A | N/A |

# G   Human evaluation details

The Google Forms survey contains 5 sections, encompassing a total of 50 questions. Instructions and examples are detailed in attached screenshots for each section.

1. **Visual Consistency in Consistent Image Generation**: Participants evaluate visual consistency across four images of the same subject, for example, "Native American sailor" produced by different methods, Methods that maintain character consistency are scored 1; others receive a score of 0. Detailed instructions are provided in Fig. 22.

2. **Text Alignment in Consistent Image Generation**: Respondents assess the alignment of text with the corresponding image for each method, assigning a score from 1 to 3, where a higher score indicates better alignment. Detailed instructions are provided in Fig. 23.

3. **Visual Diversity in Diverse Image Generation**: Like the first section, participants rate the diversity in five images of the same subject across different methods. They assign a score from 1 to 3, with a higher score indicating greater diversity. Detailed instructions are provided in Fig. 24.

4. **Text Alignment in Diverse Image Generation**: This section mirrors Section 2 but in the context of diverse image generation. Participants rate text-image alignment on a scale from 1 to 3. Detailed instructions are provided in Fig. 23.

5. **Personalized Video Quality**: Participants evaluate the quality of videos generated with the same random seed by different methods. Methods that are chosen for higher quality receive a quality score of 1; others receive a score of 0. Detailed instructions are provided in Fig. 25.

We aggregated scores from all sections and display the results in Fig. 11. For the meaning of the scores, i.e. the vertical axes in Fig. 11: we asked the participants to rate the consistency, diversity, etc., according to the rules outlined above. The value on the vertical axis represents the summation of the scores across all participants and questions.

For consistent image generation, we included 5 images per question per method. Four images evaluated subject consistency, and one image evaluated text alignment. There were 10 questions and 3 methods to compare, totaling 150 images. For diverse image generation, we followed a similar approach: 5 images per question per method, 10 questions, and 3 methods, totaling 150 images. For video generation, we included 10 videos and 2 methods, totaling 20 videos. **In total, each participant provided 140 ratings, resulting in 6,160 ratings from 44 participants.**

The results used in the human evaluation did not apply the techniques for mitigating spatial layout and background leakage described in Sec. 4.1.1. Nevertheless, our method is still preferred by the evaluators.

# H  Licenses

Pretrained models:

- ProtoVision-XL[3] [43] CreativeML Open RAIL++-M License
- Stable-Video-Diffusion-img2vid-xt-1-1[4] [5] CreativeML Open RAIL++-M License
- FLUX-dev[5] FLUX.1 [dev] Non-Commercial License
- BLIP Diffusion[6] [33] Apache 2.0 License
- IP-Adapter-SDXL[7] [67] Apache 2.0 License
- InstantID[8] [59] Apache 2.0 License

Codebase:

- diffusers 0.25.1 [9] [58] Apache 2.0 License
- EvalCrafter [10] [36] No license found

Metric models:

- OpenCLIP[11] [44, 26] MIT License
- DreamSim[12] [13] MIT License
- LPIPS 1.0[13] [68] BSD-2-Clause license

---

[3] https://huggingface.co/stablediffusionapi/protovision-xl-high-fidel
[4] https://huggingface.co/stabilityai/stable-video-diffusion-img2vid-xt-1-1
[5] https://huggingface.co/black-forest-labs/FLUX.1-dev
[6] https://huggingface.co/salesforce/blipdiffusion
[7] https://huggingface.co/h94/IP-Adapter/blob/main/sdxl_models/ip-adapter_sdxl.bin
[8] https://huggingface.co/InstantX/InstantID
[9] https://github.com/huggingface/diffusers
[10] https://github.com/EvalCrafter/EvalCrafter
[11] https://huggingface.co/laion/CLIP-ViT-g-14-laion2B-s12B-b42K
[12] https://dreamsim-nights.github.io/
[13] https://lightning.ai/docs/torchmetrics/stable/image/learned_perceptual_image_patch_similarity.html

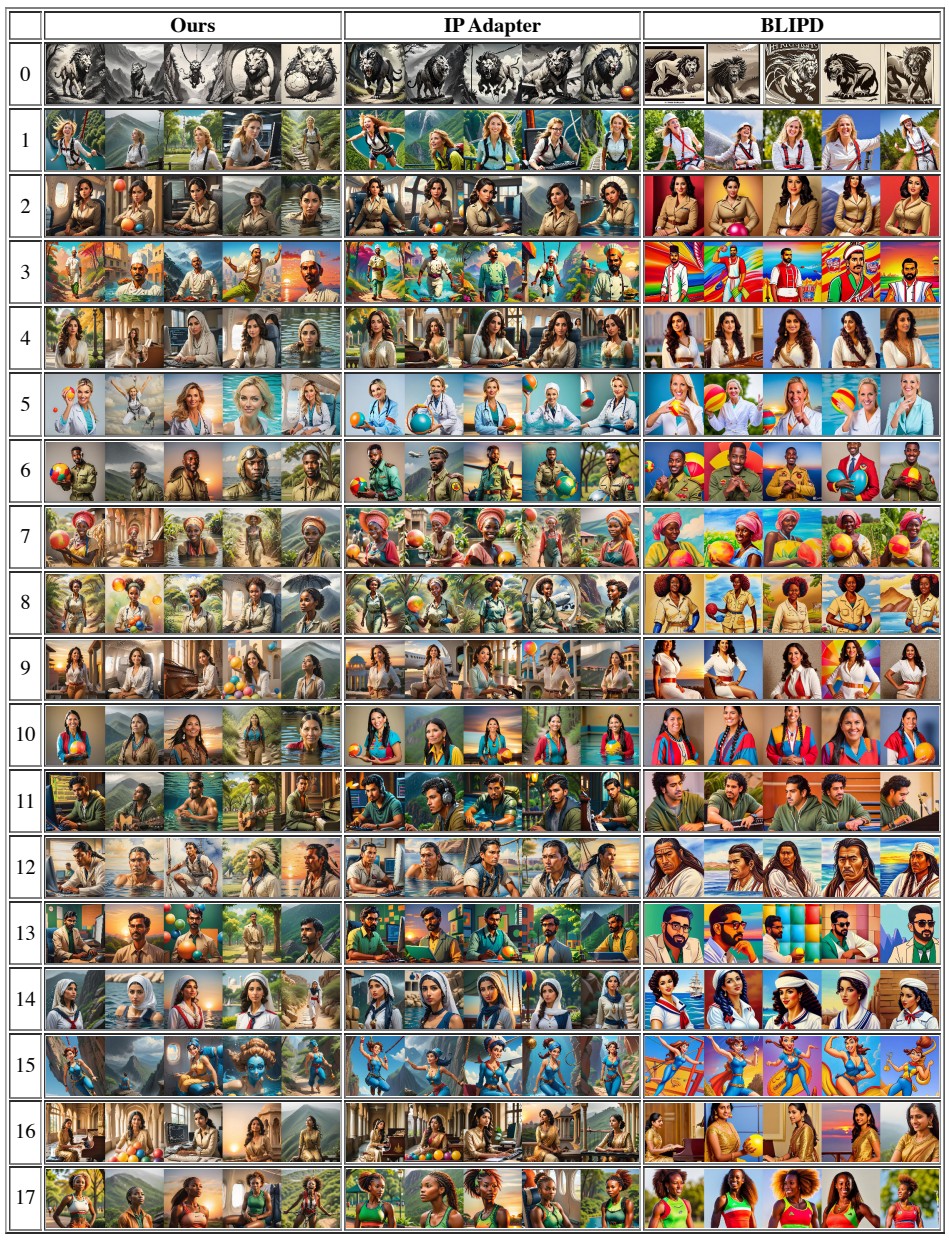

Figure 16: More consistent image generation comparison.

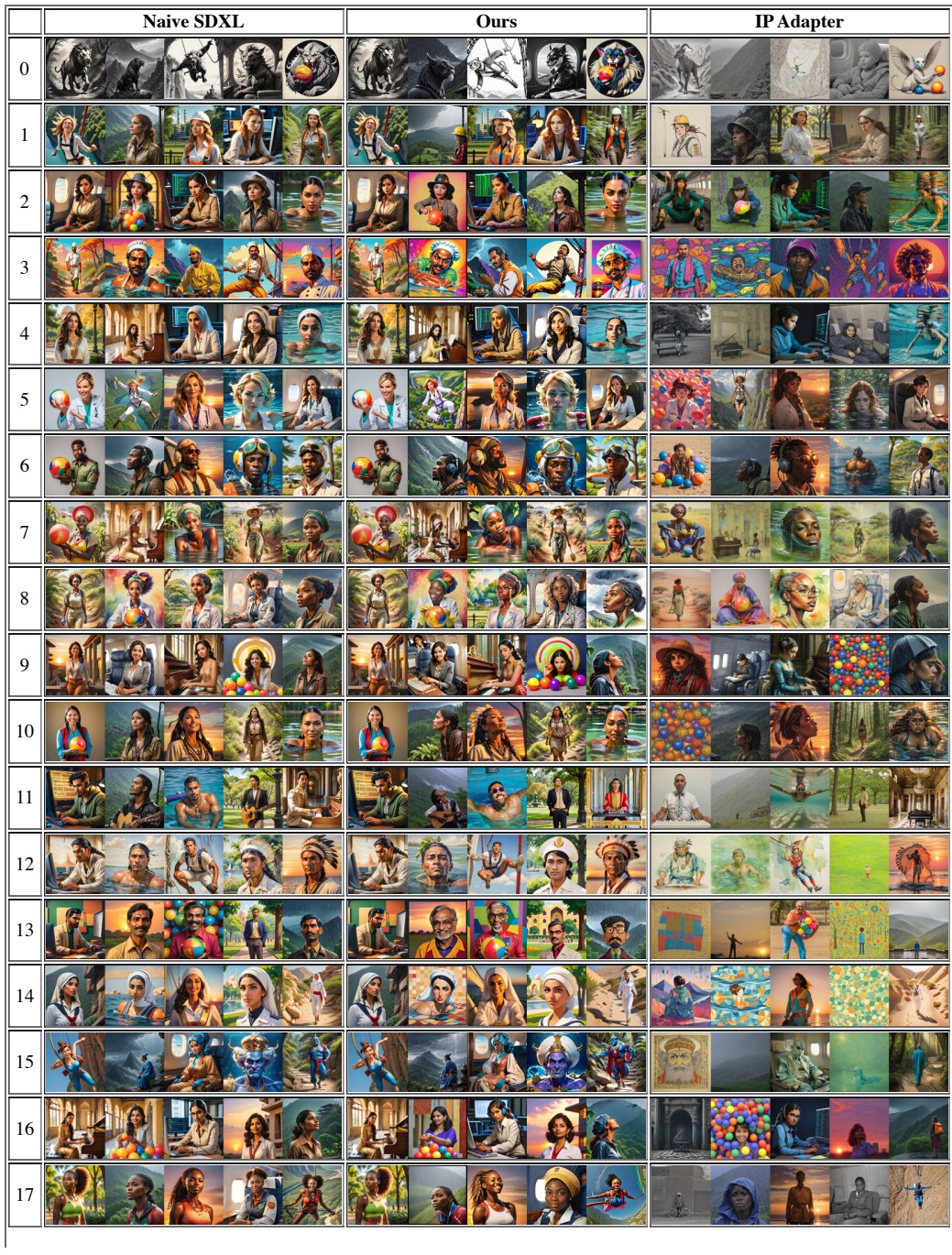

Figure 17: More diverse image generation comparison.

a smiling chimera of Russian Blue cat and Border Collie is swimming / jumping / playing guitar

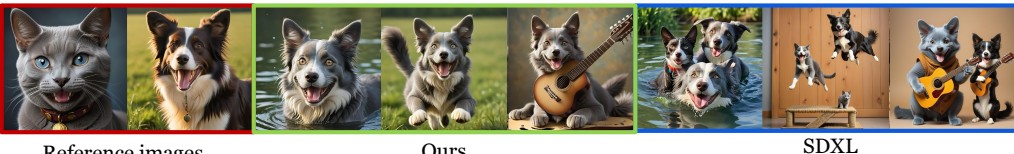

Reference images      Ours      SDXL

Figure 18: Blending a dog and a cat in various activities: `RefDrop` successfully combines features from two reference images and closely follows the text prompt, whereas SDXL struggles to generate a single cohesive object even with the guidance from the text prompt.

Blend cat and dog              Blend panda and dog

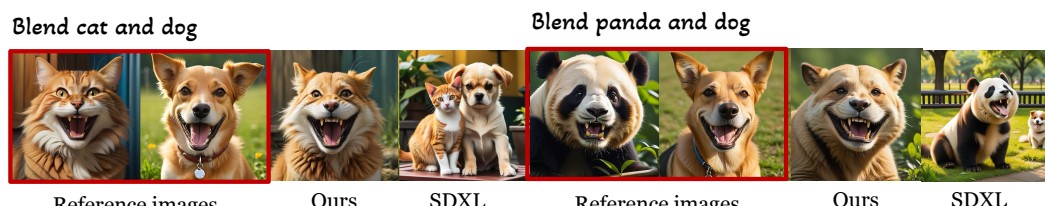

Reference images    Ours    SDXL      Reference images    Ours    SDXL

Figure 19: More visualizations for blending two distinct animals. One crucial strategy for our method to effectively blend two objects is to avoid explicitly naming them in the text prompt. We have discovered that using a generic term like "an animal" leads to better results than specifying "a cat-like dog." This trick minimizes the overly strong influence that explicit names can have, facilitating a more effective merger of the two subjects. For SDXL, we use the prompt "a chimera of [animal A] and [animal B]", but it fails to generate a single and cohesive entity.

Blend dwarf, Black widow, and Winnie the Pooh

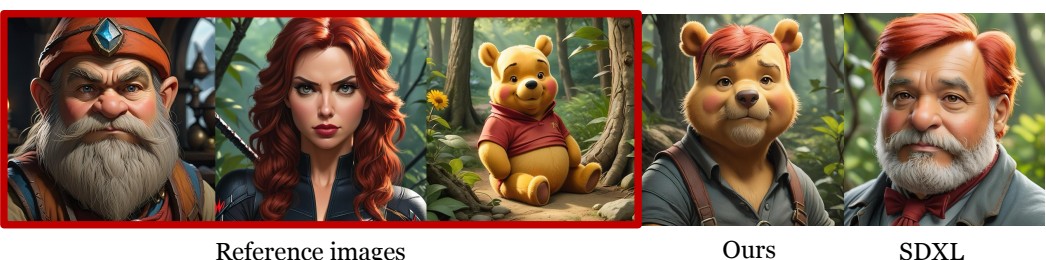

Reference images      Ours      SDXL

Figure 20: Blending **three** distinct subjects, we use the same prompt—"a portrait of Winnie the Pooh with red hair and a gray beard"—for both SDXL and `RefDrop`. However, SDXL significantly downplays the features of Winnie the Pooh. In contrast, our approach effectively absorbs the features from the reference images, retaining the dwarf's outfit and beard, Black Widow's red hair, and Winnie's facial structure.

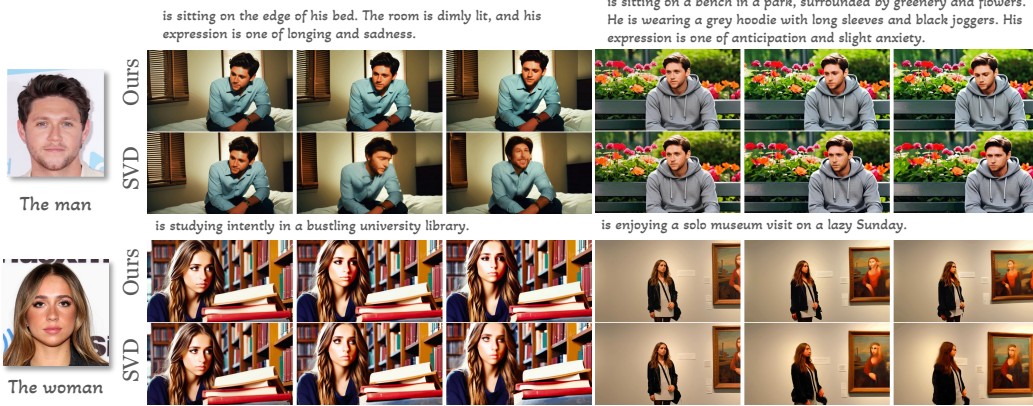

Figure 21: Additional personalized video comparison. The original videos can be viewed here.

Which group of images contain the same character? (At least choose one, multiple choices allowed)    *

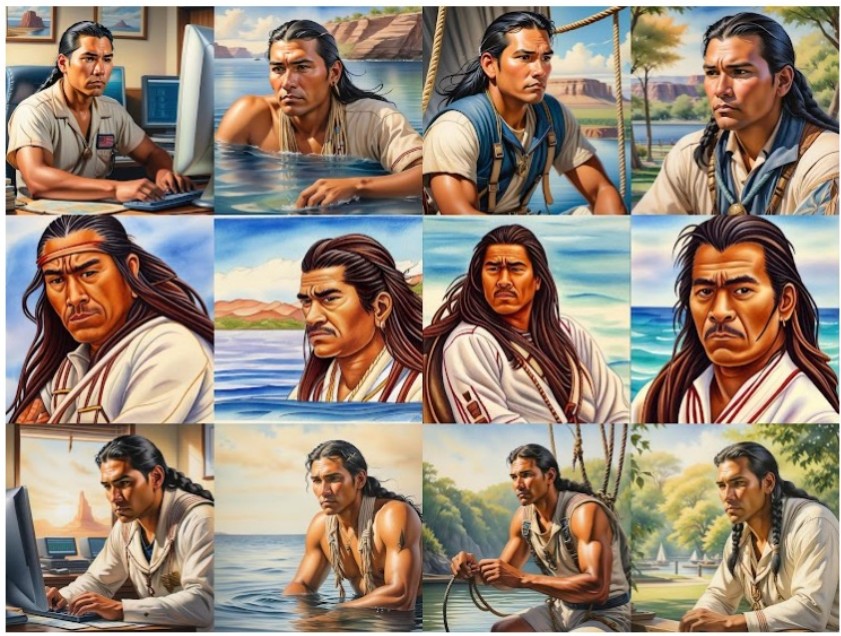

☑ Row1

☐ Row2

☐ Row3

Figure 22: The instruction and example for human evaluation.

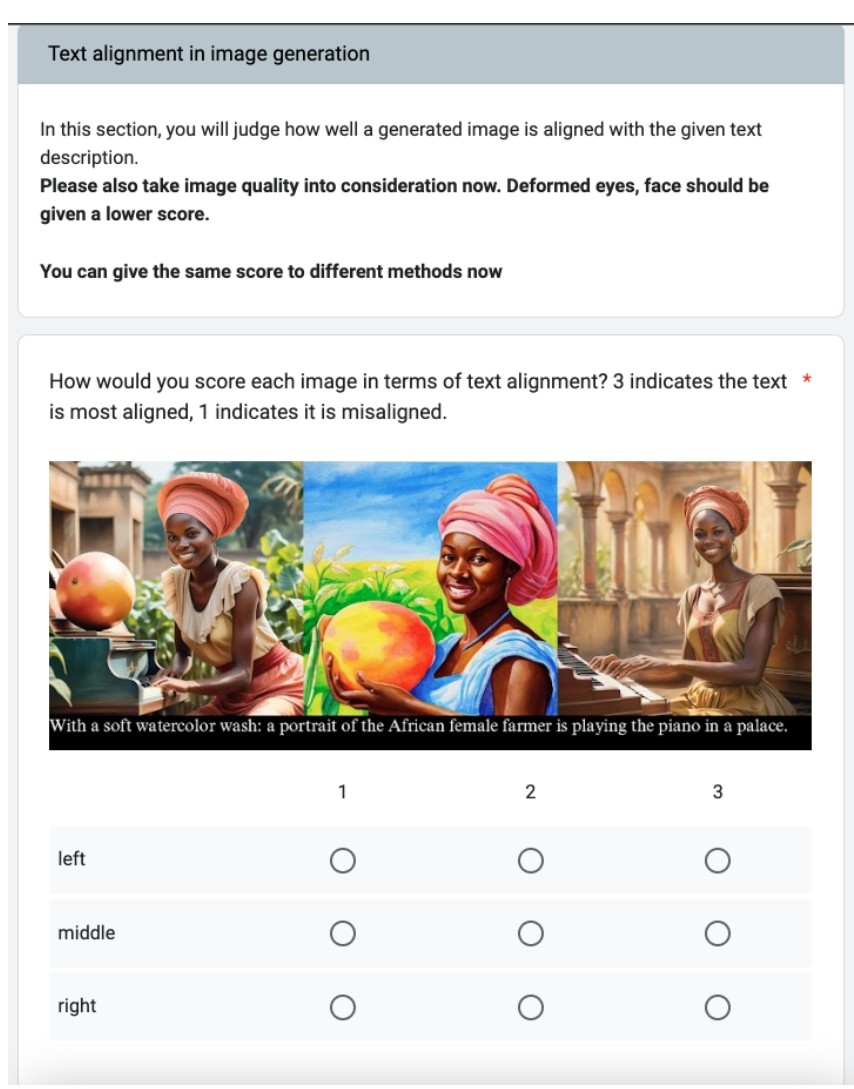

Figure 23: The instruction and example for human evaluation.

- **human: pose, accessories, hair style, age;**
- **with or without human;**
- **background, camera angle, camera focus length, image style**

For example, if a group of images can generate different accessories, such as eyeglasses, earphones, hats, that it is considered more diverse

In this section, **you need to give a rank, you cannot choose tied ranking**

Which row of images contains the most diverse content? Assign a score of 3 for * the most diversity and a score of 1 for the least.

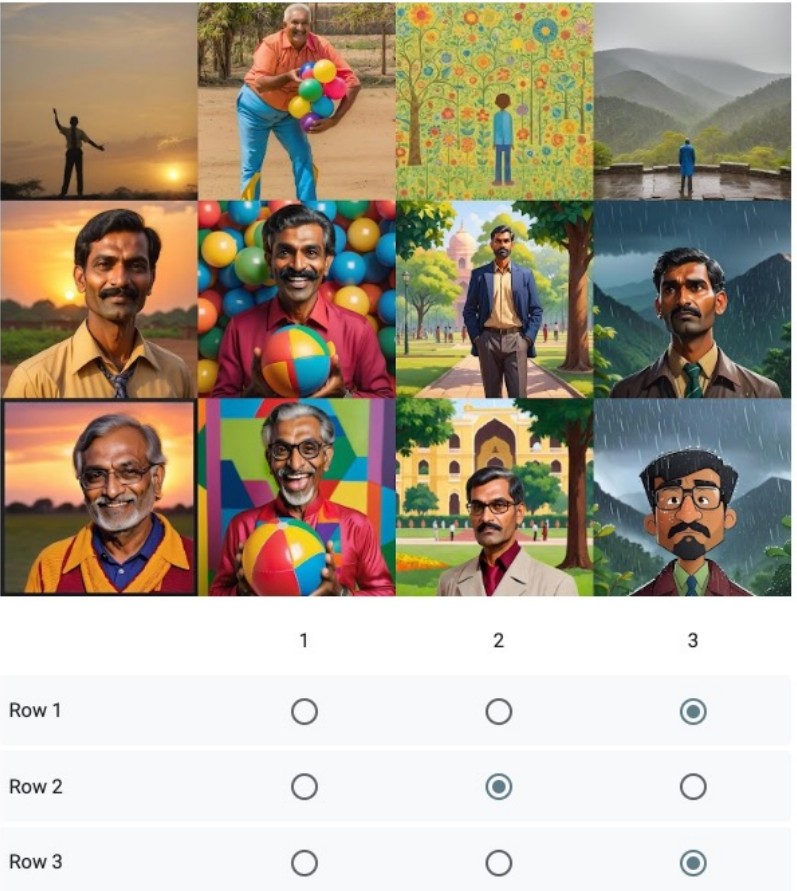

|        | 1 | 2 | 3 |
|--------|---|---|---|
| Row 1  | ○ | ○ | ◉ |
| Row 2  | ○ | ◉ | ○ |
| Row 3  | ○ | ○ | ◉ |

Figure 24: The instruction and example for human evaluation.

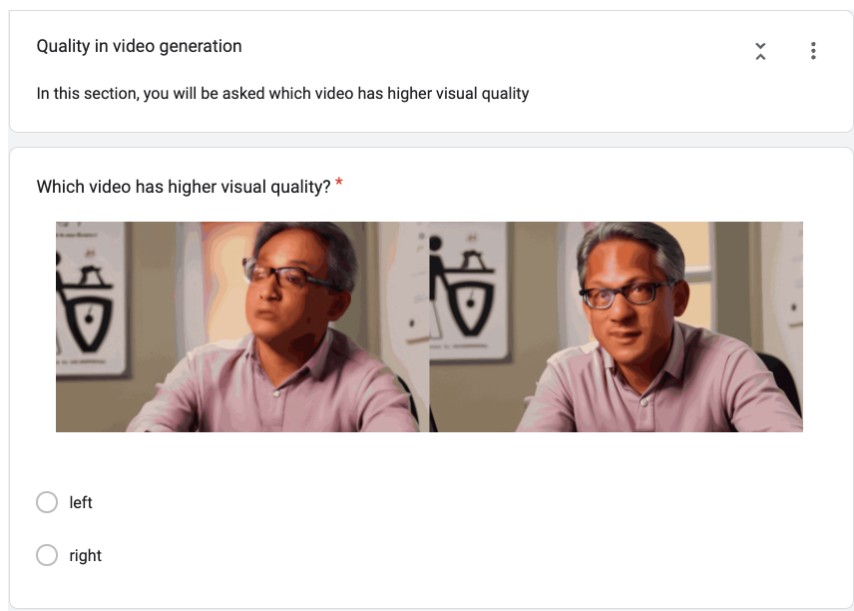

Figure 25: The instruction and example for human evaluation.

