# OpenReview forum: "RefDrop: Controllable Consistency in Image or Video Generation via Reference Feature Guidance"
_NeurIPS.cc/2024/Conference — NeurIPS 2024 poster_

### Official Review · Reviewer_Z675 · 2024-07-05

**Soundness:** 3
**Presentation:** 4
**Contribution:** 3
**Rating:** 5
**Confidence:** 4

**Summary:**

This paper targets the task of subject consistency in image and video generation. Previous consistency generation methods [53,68] are based on concatenated attention. The authors reformulate concatenated attention in a manner similar to classifier-free guidance, simplifying the constant C matrix to a constant c, leading to the proposed reference feature guidance (RFG). The authors applied RFG to multiple applications, including subject consistency in image and video generation. Qualitative and quantitative results show that RFG outperforms prior works such as IP-Adapter, BLIPD, and SDXL.

**Strengths:**

1. The authors propose a simple yet effective method for consistent text-to-image generation. The relationship among the proposed method, cross-frame attention, and concatenated attention is an interesting observation.

2. The proposed method is shown to be useful in multiple image and video generation applications, including blending features of multiple images, using negative examples to increase diversity, improving temporal consistency in video generation, and preserving identity in video generation. This method is shown to be generalizable to different techniques.

3. The paper includes comprehensive visual examples demonstrating good visual quality.

**Weaknesses:**

1. The generated images appear to have limited diversity. In Figure 13, many examples of humans have very similar poses and styles. In contrast, IP-Adapter generates more diverse images.

2. The paper discusses the relationship between concatenated attention and cross-frame attention, mentioning that the results have similar effects to recent works [53] and [68]. However, visual comparisons are not included.

3. Most results focus on humans. It would be beneficial to see different types of subjects/objects.

4. Although the proposed method successfully generates videos with temporal consistency, the generated videos contain less motion. Could the authors explain the possible reason for this and suggest ways to improve it?

**Questions:**

The text alignment decreases in Figure 14. Could the authors provide comments on why this happens and suggest possible solutions to improve text alignment?

**Limitations:**

Yes, the authors have addressed the limitations and potential negative impact.

---

> ### Author Rebuttal · Authors · 2024-08-06
>
> Thank you for your thorough comments! They are quite insightful!
>
> > The generated images appear to have limited diversity. In Figure 13, many examples of humans have very similar poses and styles.
> >
>
> Please see the global response for this matter. In short, we propose to exclude the first upsampling block from the modified attention blocks to solve this issue.
>
> > The paper discusses the relationship between concatenated attention and cross-frame attention, mentioning that the results have similar effects to recent works [53] and [68]. However, visual comparisons are not included.
> >
>
> For concatenated attention, please see the global response. Regarding cross-frame attention, we did not claim to have similar effects. Indeed, cross-frame attention should not be used in consistent image generation because it replaces the entire generated image feature with the reference image feature, which would significantly jeopardize text alignment. We compare with them in consistent video generation in *Figure* **8** of the submission and mention the static video effect on line 248.
>
> Finally, we highlight that these two attention methods have not been applied in “blender image” or “diverse image generation” tasks, while one of our novelties is that we verify our method on those tasks.
>
> > Most results focus on humans. It would be beneficial to see different types of subjects/objects.
> >
>
> We do include non-human subjects in *Figure* **1** (multi-subject consistency - dog), *Figure* **3** (cream bottle, hedgehog), *Figure* **4** (Guinea pig), and *Figures* **5**, **15**, **16**, and **17** (buildings and animals). However, we did not show any non-human subjects in diverse image generation and improving video consistency tasks. To address this, we have attached *Figures* **4** and **5** in the rebuttal PDF.
>
> > Although the proposed method successfully generates videos with temporal consistency, the generated videos contain less motion. Could the authors explain the possible reason for this and suggest ways to improve it?
> >
>
> We acknowledge this limitation as our method is training-free and plug-and-play. A straightforward way to alleviate this issue is to adjust the RFG coefficient depending on the case. To fully address this issue, we believe introducing additional training-based modules, for example, a motion guidance module proposed in StoryDiffusion [1], could provide stronger motion guidance. Alternatively, we could add motion frames as conditional guidance, as done in EMO [2].
>
> > The text alignment decreases in Figure 14. Could the authors provide comments on why this happens and suggest possible solutions to improve text alignment?
> >
>
> We believe this issue arises due to a training and testing gap. During training, the self-attention block was never trained to receive features from other reference images. Indeed, *Figure* **14** in the submission shows that some subjects are not properly generated.
>
> One straightforward solution is to use a reference coefficient with a smaller magnitude, i.e., increasing the negative coefficient $c$ from -0.3 to -0.2, with a bit sacrifice in subject diversity. Additionally, inspired by the method to solve the human pose issue, we propose excluding the first upsampling block. In the right part of *Figure* **3** in the rebuttal, we find that applying this technique improves text alignment, albeit with a slight sacrifice in subject diversity. The authors of B-Lora [3] also found that this block dominates content generation, in contrast to style or color generation. We believe removing this block can help retain the original generation’s semantic content while allowing the reference image to negatively influence style generation.
>
> **References**
>
> [1] Yupeng Zhou, Daquan Zhou, Ming-Ming Cheng, Jiashi Feng, and Qibin Hou. StoryDiffusion: Consistent self-attention for long-range image and video generation, (2024).
>
> [2] Linrui Tian, Wang Qi, Zhang Bang, and Liefeng Bo. "Emo: Emote portrait alive-generating expressive portrait videos with audio2video diffusion model under weak conditions." *arXiv preprint arXiv:2402.17485* (2024).
>
> [3] Frenkel, Yarden, Yael Vinker, Ariel Shamir, and Daniel Cohen-Or. "Implicit Style-Content Separation using B-LoRA." *arXiv preprint arXiv:2403.14572* (2024).

---

> > ### Comment · Reviewer_Z675 · 2024-08-14
> >
> > Many thanks to the authors for their detailed reply and efforts. My concerns have been well addressed.

---

### Official Review · Reviewer_c2sE · 2024-07-12

**Soundness:** 3
**Presentation:** 4
**Contribution:** 3
**Rating:** 6
**Confidence:** 4

**Summary:**

The authors analyzed the current methods of consistent content generation and revealed that the concatenation of reference features in the self-attention block can be reformulated as a linear interpolation of image self-attention and cross-attention between synthesized content and reference features with a constant rank-1 coefficient. Motivated by the analysis, the paper identified the rank-1 coefficient is not necessary and proposes a simplified method by replacing the rank-1 coefficient with a scalar coefficient. Compared to the previous method, this method doesn’t require additional training, while achieving considerable performance, and allowing multiple applications, such as multi-references, concept suppression, and video generation.

**Strengths:**

1. The method and related work are clear and well-written, easy to follow.
2. The proposed method is simplified based on previous methods and achieves effective results.
3. The method is training-free and the authors claim that it’s flexible and plug-and-play for diverse text-to-image models.
4. The paper exhibited many qualitative results and highlighted its effectiveness with comprehensive experiments for different applications.

**Weaknesses:**

1. As the paper mentions the proposed method is plug-and-play, if a subsection is added to show the flexibility of the method across different models and potential limitations/considerations when applying it, that would be great.
2. The quantitative comparisons can be difficult for this task but is it possible to use CLIP or other models to evaluate the consistency of the reference and generated images quantitatively?
3. A more comprehensive discussion on its limitations, failure cases, or social impact would be great.

**Questions:**

1. How many generated images are used for human evaluation, and what do the values for the vertical axis (such as over 300 for consistency score) mean?

2. It seems that the major difference between the proposed method and the previous method is whether to use a rank-1 matrix or a scalar for the interpolation of attention. Could the authors elaborate more on why the previous method requires training while the proposed method does not? How is the rank-1 matrix obtained in the previous method?

3. How to align the parameters for the methods using the rank-1 matrix and the scalar so that the control of the reference is comparable across methods?

**Limitations:**

The authors provide a concise discussion of limitations and it would be better to include a more comprehensive one and provide discussion of potential societal impact.

---

> ### Author Rebuttal · Authors · 2024-08-06
>
> Thank you for your review and detailed questions!
>
> > If a subsection is added to show the flexibility of the method across different models and potential limitations/considerations when applying it, that would be great.
> >
>
> We try our method on the most recent capable open-source model, FLUX-dev, developed by Black Forest Lab. We have added a figure to show the effect of Reference Feature Guidance on the FLUX-dev model in *Figure* **7** in the rebuttal PDF. Since FLUX is a transformer-based diffusion model, it does not have separate cross-attention and self-attention layers. we have to do some modification to adapt to their model. Denote $X'$ is the attention output for the $i$-th sample in one batch. In FLUX-dev, it is calculated as:
>
> $$
> X' = \text{Attention}([Q_{\text{img }i}; Q_{\text{txt }i}], [K_{\text{img }i}; K_{\text{txt }i}], [V_{\text{img }i}; V_{\text{txt }i}])
> $$
>
> where $[\cdot ; \cdot]$ denotes the concatenation operation. Similar to Equation (4), we modify the attention blocks in FLUX to accept additional guidance from the reference image feature "img $1$", assumed to be the first image in the batch:
>
> $$
> \begin{align*}
> X_{\texttt{RFG}}'
> = X' + c \cdot (\text{Attention}(Q_{\text{img } i}, K_{\text{img } 1}, V_{\text{img } 1}) - \text{Attention}(Q_{\text{img } i}, K_{\text{img } i}, V_{\text{img } i}))
> \end{align*}
> $$
>
> We add two additional terms multiplied by a guidance scale $c$ on top of the original attention output $X'$. And those additional terms only depend on the image features, ensuring that we do not interfere with the text features "txt $i$". *Figure* **7** in the rebuttal PDF demonstrates that our method effectively applies reference guidance to the FLUX generation.
>
> Potential limitations when applying RefDrop to other models:
>
> Firstly, we find that the coefficient parameter $c$ may require additional tuning. Next, for the technique we proposed to solve spatial layout leakage, an ablation study is needed to determine which layer dominates spatial layout generation, as this layer may vary across different models.
>
> > is it possible to use CLIP or other models to evaluate...?
> >
>
> For image generation, we conducted a quantitative study in *Figure* **12** of Section C.1 of the submission. We used the OpenCLIP model to measure text alignment, DreamSim to measure subject similarity, and LPIPS to measure subject diversity, following Consistory [1]. Unfortunately, we made a mistake in the text similarity calculation, so *Figure* **12** is not reliable. We corrected the error and redid the quantitative study, presenting the results in *Figure* **3** of the rebuttal PDF. Please also see the global response for clarification on this part. *Figure* **3** in the rebuttal shows that our method achieves a good balance between text alignment and subject consistency/diversity.
> For video generation, we already have a quantitative study in Table 2 in the paper appendix.
>
> > How many generated images are used for human evaluation, and what do the values for the vertical axis (such as over 300 for consistency score) mean?
> >
>
> For consistent image generation, we included 5 images per question per method. Four images evaluated subject consistency, and one image evaluated text alignment. There were 10 questions and 3 methods to compare, totaling 150 images.
>
> For diverse image generation, we followed a similar approach: 5 images per question per method, 10 questions, and 3 methods, totaling 150 images.
>
> For video generation, we included 10 videos and 2 methods, totaling 20 videos.
>
> For the meaning of the scores: we asked the participants to rate the consistency, diversity, etc., according to the rules outlined in Section F of the submission appendix. The value on the vertical axis represents the summation of the scores across all participants and questions.
>
> > It seems that the major difference between the proposed method and the previous method is whether to use a rank-1 matrix or a scalar for the interpolation of attention. Could the authors elaborate more on why the previous method requires training while the proposed method does not?
> >
>
> We want to clarify that by the rank-1 matrix method, we mean the concatenated attention method, which is also training-free. Consistory uses this method and is a training-free method. Please see Table 1 in the paper submission for reference.
>
> > How is the rank-1 matrix obtained in the previous method?
> >
>
> The rank-1 matrix used in the concatenated attention-based method is not explicitly shown because they use Equation (2). One of our contributions is demonstrating that Equation (2) can be equivalently reformulated as Equation (3), where the rank-1 matrix appears as the coefficient. We further simplify Equation (3) by changing the rank-1 matrix to a scalar. We emphasize that in the concatenated attention-based method, the rank-1 matrix is not manually adjustable or defined by the user; it is intrinsically determined by the reference image feature and generated image feature.
>
> > How to align the parameters for the methods using the rank-1 matrix and the scalar...?
> >
>
> We emphasize that the concatenated attention method can fall into our framework using a rank-1 matrix as the coefficient. This equivalent relationship is elaborated in Section B of the submission appendix. If we only use a scalar as the coefficient, we cannot exactly replicate the concatenated attention method. However, we empirically find that using a coefficient of 0.3 ~ 0.4 can produce a similar effect to the concatenated attention method. Please refer to the second point of the global response and *Figure* **6** in the rebuttal PDF.

---

> ### Author Response · Authors · 2024-08-07
> **Continue the rebuttal**
>
> > A more comprehensive discussion on its limitations, failure cases, or social impact would be great.
> >
> Limitations:
>
> - For consistent image generation, our method sometimes fails to replicate the exact hairstyle and outfit from the reference image. This could be a critical requirement for tasks like advertisement video production.
> - For diverse image generation, our method currently cannot precisely control aspects of diversity, such as generating diverse subjects but not styles, or vice versa. Our method does not disentangle these two aspects.
> - For blending multiple images, the coefficient needs careful tuning to achieve a good balance among multiple reference images, and it heavily depends on the specific case.
> - For improving video temporal consistency, as noted by reviewer **Z675**, although our method successfully generates videos with temporal consistency, the generated videos contain less motion.
>
> Potential societal impact: The proposed method has several potential societal impacts across various domains:
>
> - For media and advertising, it can enhance the design of key frames in advertisements or movie videos, allowing companies to create more visually compelling content.
> - Individual artists may leverage this technique to blend multiple images, potentially sparking new creative inspirations.
> - RefDrop could be used to improve temporal consistency in short videos, which may be particularly valuable for social media platforms seeking to enhance AI-generated video content.

---

> ### Comment · Reviewer_c2sE · 2024-08-11
>
> Thanks for the detailed information from the authors. My concerns are addressed and I will increase my rating from 5 to 6. Please make sure to include the clarified information in the revised paper.

---

> > ### Author Response · Authors · 2024-08-12
> >
> > Thank you very much for taking the time to read our rebuttal and raising the score! We will include the clarified information in the revised paper.

---

### Official Review · Reviewer_NJob · 2024-07-12

**Soundness:** 3
**Presentation:** 3
**Contribution:** 3
**Rating:** 5
**Confidence:** 4

**Summary:**

This paper focuses on the challenge of ensuring consistency in the generation of images and videos. Deep learning and artificial intelligence techniques are utilized in image and video generation for generating new images or generate video frames based on given inputs, such as text prompts or reference images. Ensuring consistency is necessary for maintaining coherence between the subject or object in the generated images or video frames.

The previous methods mainly consisted of the IP (image prompt)-Adapter and concatenated attention approaches. The IP-Adapter technique retrieves characteristics from reference images and incorporates them into the generation process to improve coherence. More precisely, the method employs an image encoder that has been trained independently to extract features from the reference image. These features are then incorporated into the generation process using attention mechanisms, which involve both self-attention and cross-attention.
Nevertheless, this method necessitated distinct training of the encoder and utilized unchanging weights, leading to restricted adaptability. Furthermore, it is difficult to maintain the coherence between the textual content and the accompanying visual references.

This paper introduces a novel approach called RefDrop to address these challenges. The major contributions are as follows:
1.	A review of methods for generating consistency. The paper demonstrates that current concatenated attention methods can be understood as linear combinations of self-attention and cross-attention. This discovery indicates these techniques merge features from reference images using linear interpolation.
2.	Reference Feature Guidance (RFG): The RFG method is presented as an approach for directly managing the impact of reference images by linearly combining self-attention and cross-attention. Users can enhance adaptability by adjusting the influence of reference images using a scalar coefficient 𝑐.
3.	No prior training necessary: RefDrop provides an adaptable method to efficiently utilize features from reference images without requiring separate training of the encoder.

RefDrop is a novel method that utilizes features from reference images to effectively control consistency in the generation of images and videos. This method controls the impact of reference images by using linear interpolation of self-attention and cross-attention. More precisely, features are derived from both the input and reference images, and subsequently, self-attention and cross-attention are executed independently. The outcomes of these considerations are subsequently combined through linear interpolation to generate the ultimate features, employing a scalar coefficient c to equitably distribute the influences from each attention mechanism. This enables adaptable manipulation of the impact of reference images. RefDrop efficiently employs reference features without requiring additional training, making it suitable for a wide range of generation tasks. Furthermore, it guarantees temporal consistency in video generation by utilizing the initial frame as a reference image, thereby preserving coherence throughout the following frames.

**Strengths:**

- Integration of RefDrop into Diffusion Models Without Additional Training
- Controlling Reference Image Impact in RefDrop for Efficient Image Generation
- High-Quality Image Generation with RefDrop Using Single and Multiple References
- Temporal Consistency in Video Generation with RefDrop

**Weaknesses:**

- The lack of feature-based comparisons using metrics like Kullback-Leibler (KL) divergence in the experiments

**Questions:**

This approach may introduce undesired background elements, complicating the management of specific image components, isn't it?

**Limitations:**

- The lack of attention between the reference images and text, which can result in feature entanglement

---

> ### Author Rebuttal · Authors · 2024-08-06
>
> Thank you for your review!
>
> > The lack of feature-based comparisons using metrics like Kullback-Leibler (KL) divergence in the experiments.
> >
>
> KL divergence measures the divergence between two distributions, which would require at least hundreds of images in our case. However, we aim to measure the consistency or diversity between two individual images. Therefore, KL divergence is not an appropriate metric for our purposes.
>
> > This approach may introduce undesired background elements, complicating the management of specific image components, isn't it?
> >
>
> In practice, we find that as long as we use a good text-to-image model with strong prompt adherence, such as ProtoVision-XL, background leakage is not a significant issue because the prompt adherence dominates the generation. Occasionally, background leakage can occur, and we propose using a subject mask to address this issue, as done in Consistency. Please see the global response for further clarification and *Figure* **2** in the rebuttal PDF for visual support.
>
> > The lack of attention between the reference images and text, which can result in feature entanglement.
> >
>
> We intentionally do not include attention between the reference image ($I_1$) and the current generation ($I_i$) text because we want to keep the reference image's diffusion process intact. We are not entirely sure about your question, if you could provide more details, that would be helpful for us to answer!

---

> > ### Comment · Reviewer_NJob · 2024-08-14
> >
> > Many thanks for your feedback.
> > 1) I understand what the authors did for quantitative evaluation.
> > 2) To demonstrate the reason of less background leakage, I think that visualizing the attention output can be one of solutions.

---

### Official Review · Reviewer_5p1r · 2024-07-12

**Soundness:** 2
**Presentation:** 3
**Contribution:** 2
**Rating:** 6
**Confidence:** 4

**Summary:**

This paper presents RefDrop, a method that allows users to control the influence of reference context in a direct and precise manner. More specifically, the proposed method is training-free, which means it can be used plug-and-play without the need to train a separate image encoder for feature injection from reference images.

**Strengths:**

- The paper is well written and easy to follow.
- This paper presents an intuitive way for consistent image/video generation by fusing the features of objects into the diffusion process.
- Experiments show the effectiveness of proposed method on preserving object appearance.

**Weaknesses:**

- One problem of the proposed approach is that it does not dis-entangle spatial control with appearance control. A stronger guidance scale not only means higher appearance similarity, but also higher spatial similarity between the generated image/video and the reference image. This could be an intrinsic drawback of the proposed method.

**Questions:**

- Can the authors provide some gifs for generated videos? I'm a little bit concerned that if we enforce high guidance scale, the generated video will probably be quite static without large movements.
- I guess reference-ControlNet is also relevant? It would be nice to include it as a baseline.

**Limitations:**

See weakness.

---

> ### Author Rebuttal · Authors · 2024-08-06
>
> Thank you for your review!
>
> > It does not dis-entangle spatial control with appearance control.
> >
>
> Thank you for raising this issue. Please see the global response and *Figures* **1** and **2** in the rebuttal PDF for a detailed explanation. In short, we propose excluding the first upsampling block from the modified attention blocks to address this issue.
>
> > Can the authors provide some gifs for generated videos? I'm a little bit concerned that if we enforce high guidance scale, the generated video will probably be quite static without large movements.
> >
>
> Our main paper submission includes an external link in the caption of *Figure* **9** in the paper submission, with the statement: “The additional videos can be viewed here.” Please click on the word “here” to access the external link. Additionally, we also provide gifs in *Figure* **5** of the rebuttal PDF. We agree with your concern that applying a very large reference feature guidance scale could result in static videos, similar to the effects of concatenated attention or cross-frame attention shown in *Figure* **8** of the main paper. Therefore, we typically use a small guidance scale, \( c = 0.2 \), which is sufficient to reduce video flickering without causing the video to become static.
>
> > I guess reference-ControlNet is also relevant? It would be nice to include it as a baseline.
> >
>
> We have included the reference-only ControlNet of the SDXL model in *Figure* **6** of the rebuttal PDF. We want to mention several drawbacks of Reference-only ControlNet based on our observations:
>
> - Reference-only ControlNet is also subject to spatial layout issues.
> - While Reference-only ControlNet performs well on simple subjects, such as cartoon characters, it struggles to generate humans. It is likely to produces humans with distorted eyes and bodies.

---

> > ### Comment · Reviewer_5p1r · 2024-08-12
> >
> > Thanks for the authors' response! Here are one remaining question after reading your rebuttal:
> >
> > - I personally don't find it very intuitive to exclude the first upsampling block. Other than Figure 3, do you have some intuitive explanations for this? And has such idea of excluding the first upsampling block explored in previous works? You mentioned that "the first upsampling block predominantly influences the spatial layout.", do you have some quantiative analysis to prove that removing the first upsampling block is more effective than removing other blocks?
> >
> > Thanks!

---

> ### Author Response · Authors · 2024-08-12
>
> Thank you for reading our rebuttal! We are more than happy to share our thoughts on your questions.
>
> > Other than Figure 3, do you have some intuitive explanations for this?
> >
>
> Our approach is largely inspired by recent advancements in fine-grained control for style transfer using SDXL UNet [1,2,3,4]. We hypothesize that the blocks processing low-resolution image feature maps—particularly those near the bottleneck in the UNet—exert more control over the overall coarse structure or core formulation of the image. In contrast, the blocks processing high-resolution image feature maps play a crucial role in controlling high-frequency details, a task that is difficult for low-resolution blocks to manage.
>
> Regarding why the first upsampling block influences the spatial layout the most, rather than a middle or final downsampling block, we conjecture it is because the other blocks are not yet sufficiently close to the final output image to have a significant impact. We conducted several ablation studies and found empirically that excluding the first upsampling block is the most effective in generating more diverse poses.
>
> > And has such idea of excluding the first upsampling block explored in previous works?
> >
>
> We are indeed inspired by a series of works that have demonstrated how different blocks of the SDXL UNet affect the final image generation. These works include, but are not limited to, InstantStyle [1], B-Lora [2], and discussions within the open-source community [4]. Most of these studies [1, 3, 4] focus on injecting style from a reference image and investigating which blocks most influence style generation. Meanwhile, B-Lora also analyzes which blocks have the greatest impact on content generation. In particular, B-Lora identifies the first upsampling block as the most influential for content generation, significantly controlling the spatial layout—a finding that aligns with our observation.
>
> However, these studies are primarily concerned with style transfer for text-to-image models. The key difference is that they aim to preserve the spatial layout or style from the reference image as much as possible, while we seek to exclude the spatial layout information from the reference image as much as possible. **Regarding consistent generation, we have not yet observed any work that uses this method to diversify object poses.** Previous consistent generation method Consistory [5] introduced several techniques to avoid the issue of similar poses, such as self-attention dropout and using vanilla query features. We believe that our approach of excluding the first upsampling block offers a simpler and more principled solution in practice.
>
> > Do you have some quantitative analysis to prove that removing the first upsampling block is more effective than removing other blocks?
> >
>
> We also include the quantitative results here. Specifically, we conducted 11 groups of experiments, where in each group, we excluded one of the 11 blocks of the UNet from applying Reference Feature Guidance. And in each group, we generated 20 sets of consistent objects, with each set containing 5 images. This resulted in a total of $11\times 20 \times 5 = 1100$ images for metric calculation. We then used `DreamSim` and `LPIPS` to measure the distance between the generated images and the reference image, and report the mean and standard deviation. Higher values of these metrics should indicate more diverse poses, as these scores tend to favor similar layouts [5]. From the table, we can see that excluding the first upsampling block greatly boosts the spatial layout diversity.
>
> | Excluded block | DreamSim distance | LPIPS distance |
> | --- | --- | --- |
> | Down1 | 0.2283 ± 0.0102 | 0.5484 ± 0.0052 |
> | Down2 | 0.2261 ± 0.0118 | 0.5484 ± 0.0043 |
> | Down3 | 0.2298 ± 0.0082 | 0.5475 ± 0.0058 |
> | Down4 | 0.2332 ± 0.0095 | 0.5551 ± 0.0072 |
> | Mid | 0.2484 ± 0.0048 | 0.5673 ± 0.0070 |
> | Up1 | **0.3077 ± 0.0092** | **0.5880 ± 0.0047** |
> | Up2 | 0.2567 ± 0.0087 | 0.5603 ± 0.0077 |
> | Up3 | 0.2440 ± 0.0088 | 0.5580 ± 0.0032 |
> | Up4 | 0.2441 ± 0.0079 | 0.5611 ± 0.0074 |
> | Up5 | 0.2368 ± 0.0106 | 0.5540 ± 0.0055 |
> | Up6 | 0.2455 ± 0.0103 | 0.5639 ± 0.0075 |
>
> **Reference**
>
> [1] Wang, Haofan, Qixun Wang, Xu Bai, Zekui Qin, and Anthony Chen. "Instantstyle: Free lunch towards style-preserving in text-to-image generation." *arXiv preprint arXiv:2404.02733* (2024).
>
> [2] Frenkel, Yarden, Yael Vinker, Ariel Shamir, and Daniel Cohen-Or. "Implicit Style-Content Separation using B-LoRA." *arXiv preprint arXiv:2403.14572* (2024).
>
> [3] Jeong, Jaeseok, Junho Kim, Yunjey Choi, Gayoung Lee, and Youngjung Uh. "Visual Style Prompting with Swapping Self-Attention." *arXiv preprint arXiv:2402.12974* (2024).
>
> [4] https://github.com/huggingface/diffusers/discussions/7534
>
> [5] Yoad Tewel, Omri Kaduri, Rinon Gal, Yoni Kasten, Lior Wolf, Gal Chechik, and Yuval Atzmon. Training-free consistent text-to-image generation. arXiv preprint arXiv:2402.03286, 2024.

---

> > ### Comment · Reviewer_5p1r · 2024-08-12
> >
> > Thanks for this detailed explanation from the authors, my question has been well-addressed, really appreciate it.
> > I will increase my rating from 4 to 6. Please make sure to include these discussions in the revised version, since intuitions and observations sometimes is more useful for other researchers.

---

> > > ### Author Response · Authors · 2024-08-13
> > >
> > > Thank you very much for your response! We quite enjoy this rebuttal process because of your insightful feedbacks, and we are already working on including these discussions in the revised version.

---

### Author Rebuttal · Authors · 2024-08-06

We thank the reviewers for their valuable comments. We are pleased that the reviewers find our method intuitive (Reviewer 5p1r) and plug-and-play (Reviewer NJob, c2sE), and that they consider our paper well-written (Reviewers 5p1r, c2sE). We appreciate the acknowledgment from all reviewers that our experimental results comprehensively demonstrate the effectiveness of our method. We believe our RefDrop method will advances the controllable consistency for image and video generation. Below, we address the reviewers’ common concerns:

---

**Redundant Information Leakage in Consistent Image Generation**

The primary criticism (**Z675, NJob, 5p1r**) concerns redundant information leakage from the reference image in consistent image generation, specifically in two areas:

1. **Spatial Layout Leakage (Z675, 5p1r)**: This issue can result in generated objects having similar poses. SDXL UNet consists of 4 downsampling blocks, 1 middle block, and 6 upsampling blocks. Initially, we applied Reference Feature Guidance (RFG) to all 11 attention blocks. Through an ablation study, we found that the first upsampling block predominantly influences the spatial layout. By excluding this block from the modified attention blocks, we were able to recover diverse human poses. *Figures* **1** and *2* in the rebuttal PDF visually illustrate our observations.
2. **Background Leakage (NJob)**: This issue can cause the generated images to have backgrounds very similar to the reference image. Although this issue is less significant, we applied a subject mask to address it, as done in Consistory [1]. *Figure* **2** in the rebuttal PDF visually verifies our claim.

In the left part of *Figure* **3** in the rebuttal PDF, we have redone the quantitative experiment in the submission appendix, adding results for RefDrop with new techniques—keeping one upsampling block unchanged and adding a subject mask. The IP-Adapter tends to be overly influenced by the reference image, significantly decreasing text alignment. For example, if the reference image shows a human holding a ball, many generated images would also show a ball (see *Figure* **13** in the submission appendix). After adding new techniques, our method achieves diverse spatial poses and resolves the background issue, leading to significant improvements in text alignment. Our subject consistency score drops due to the metric used (DreamSim), which would bias its score towards configurations with similar spatial layouts or subject actions. Thus, our method now achieves a good balance between text alignment and subject consistency, without overfitting to the reference image spatial layout.

**Clarification on the Relationship with Concatenated Attention**

Another concern raised by **Z675** and **c2sE** is the comparison with concatenated attention. In *Figure* **6** of the rebuttal PDF, we present a visual comparison with concatenated attention, using the same random seed and without additional techniques such as excluding one attention block (as proposed by us) and self-attention dropout from Consistory [1]. This comparison confirms our claim in the submission: “the concatenated attention used in studies like Tewel et al. [53] and Zhou et al. [68] produces results similar to our RFG mechanism.” In practice, we find that RFG with a coefficient of 0.3~0.4 produces results quite similar to the concatenated attention method.

**Finally, we encourage all the reviewers to see our rebuttal PDF, and zoom in to see all the details! To see the gifs in the PDF, it’s suggested to use Adobe Acrobat or Foxit Reader.**

**Reference**

[1] Yoad Tewel, Omri Kaduri, Rinon Gal, Yoni Kasten, Lior Wolf, Gal Chechik, and Yuval Atzmon. Training-free consistent text-to-image generation. arXiv preprint arXiv:2402.03286, 2024.

---

### Decision · Program_Chairs · 2024-09-25

**Decision:**

Accept (poster)

**Comment:**

The reviewers acknowledge the contribution of proposing a training-free method with promising results. The rebuttal was successful, resulting in an increased average score, with previous negative scores now becoming supportive. After careful discussion and consideration, we are pleased to inform you that your paper has been accepted. However, the final version will require revisions to reflect the important discussions presented in the rebuttal. Specifically, please add quantitative results and analysis, and include a discussion addressing concerns regarding background leakage and reduced motion in the video.